# Agro-Morphological Exploration of Some Unexplored Wild *Vigna* Legumes for Domestication

**Difo Voukang Harouna** [1,2,*] **, Pavithravani B. Venkataramana** [2,3]**, Athanasia O. Matemu** [1] **and Patrick Alois Ndakidemi** [2,3]

[1] Department of Food Biotechnology and Nutritional Sciences, Nelson Mandela African Institution of Science and Technology (NM-AIST), P.O. Box 447, Arusha 23311, Tanzania; athanasia.matemu@nm-aist.ac.tz

[2] Centre for Research, Agricultural Advancement, Teaching Excellence and Sustainability in Food and Nutrition Security (CREATES-FNS), Nelson Mandela African Institution of Science and Technology, P.O. Box 447, Arusha 23311, Tanzania; pavithravani.venkataramana@nm-aist.ac.tz (P.B.V.); Patrick.ndakidemi@nm-aist.ac.tz (P.A.N.)

[3] Department of Sustainable Agriculture, Biodiversity and Ecosystems Management, Nelson Mandela African Institution of Science and Technology, P.O. Box 447, Arusha 23311, Tanzania

\* Correspondence: harounad@nm-aist.ac.tz or difovoukang@yahoo.fr

**Abstract:** The domestication of novel or hitherto wild food crops is quickly becoming one of the most popular approaches in tackling the challenges associated with sustainable food crop production, especially in this era, where producing more food with fewer resources is the need of the hour. The crop breeding community is not yet completely unanimous regarding the importance of crop neo-domestication. However, exploring the unexplored, refining unrefined traits, cultivating the uncultivated, and popularizing the unpopular remain the most adequate steps proposed by most researchers to achieve the domestication of the undomesticated for food and nutrition security. Therefore, in the same line of thought, this paper explores the agro-morphological characteristics of some wild *Vigna* legumes from an inquisitive perspective to contribute to their domestication. One hundred and sixty accessions of wild *Vigna* legumes, obtained from gene banks, were planted, following the augmented block design layout of two agro-ecological zones of Tanzania, during the 2018 and 2019 main cropping seasons for agro-morphological investigations. The generalized linear model procedure (GLM PROC), two-way analysis of variance (two-way ANOVA), agglomerative hierarchical clustering (AHC) and principal component analysis (PCA) were used to analyze the accession, block and block vs. accession effects, as well as the accession × site and accession × season interaction grouping variations among accessions. The results showed that the wild species (*Vigna racemosa*; *Vigna ambacensis*; *Vigna reticulata*; and *Vigna vexillata*) present a considerable variety of qualitative traits that singularly exist in the three studied checks (cowpea, rice bean, and a landrace of *Vigna vexillata*). Of the 15 examined quantitative traits, only the days to flowering, pods per plant, hundred seed weight and yield were affected by the growing environment (accession × site effect), while only the number of flowers per raceme and the pods per plant were affected by the cropping season (accession × season effect). All the quantitative traits showed significant differences among accessions for each site and each season. The same result was observed among the checks, except for the seed size trait. The study finally revealed three groups, in a cluster analysis and 59.61% of the best variations among the traits and accessions in PCA. Indications as to the candidate accessions favorable for domestication were also revealed. Such key preliminary information could be of the utmost importance for the domestication, breeding, and improvement of these species, since it also determines their future existence—that is, so long as biodiversity conservation continues to be a challenging concern for humanity.

**Keywords:** undomesticated legumes; *Vigna racemosa*; *Vigna ambacensis*; *Vigna reticulata*; *Vigna vexillata*; wild food legumes; legumes; *Vigna* species; domestication; unexplored legumes

## 1. Introduction

It has been reported that only 12 crops contribute most to the current global food production, with only three of them (rice, wheat and maize) providing more than 50% of the world's calories [1–3]. Consequently, the Food and Agricultural Organization of the United Nations has predicted that the world is in need of about 70% more food to adequately feed the ever-growing population [4]. Therefore, a detailed screening and exploration of hitherto wild and novel species from various agro-climatic regions of the world could help to mitigate the need. Although the crop breeding community is not yet unanimous on the importance of the crop neo-domestication concept, it is increasingly becoming popular in research on this topic that domesticating the undomesticated is an ideal method, which could aid in mitigating the global food insecurity challenges [1,5]. The method will not only promote the successful utilization of hitherto wild and non-domesticated food crops in dietary diversification programs, but also help with biodiversity conservation.

Legumes (family: Fabaceae) constitute the third largest family of flowering plants. [6]. However, only ten species have been domesticated and recognized as human food [5]. Those few domesticated ones have incontestably proven to be of crucial nutritional value for both humans and animals due to their protein content, causing them to be recognized as the second most valuable plant source of nutrients [6]. Despite their positive impact on global food and nutrition security, it has also been reported that their production rate remains unsatisfying, as compared with their consumption rate, due to biotic and abiotic challenges [7]. Therefore, there is a need to look for sustainable alternative strategies to improve and diversify their production. A systematic exploration of the hitherto wild undomesticated and wild relatives of the domesticated species within the commonly known and the little-known genera of legumes might be a hopeful strategy.

The *Vigna* genus is a large group of legumes consisting of more than 200 species, of which some are of agronomic, economic, and environmental importance [8]. The most common domesticated ones include the mung bean [*V. radiata* (L.) Wilczek], urd bean [*V. mungo* (L.) Hepper], cowpea [*V. unguiculata* (L.) Walp], azuki bean [*V. angularis* (Willd.) Ohwi and Ohashi], bambara groundnut [*V. subterranea* (L.) Verdn.], moth bean [*V. aconitifolia* (Jacq.)], and rice bean [*V. umbellata* (Thunb.) Ohwi and Ohashi]. They have recognized usages, ranging from forage, green manure, and cover crops, in addition to their high-protein grains. However, they are very limited in number, as compared with the existing wild ones.

The *Vigna* genus legumes also comprise more than 100 wild species, which do not have common names, apart from their scientific appellation [9]. They are given different denotations, such as the under-exploited wild *Vigna* species, undomesticated *Vigna* species, wild *Vigna* or alien species, depending on the scientist [5,8,10]. These constitute the main subject of interest in this research, as very little information about them, including their agro-morphological characteristics, has been reported. For the purpose of this study, accessions of *Vigna racemose*, *Vigna ambacensis*; *Vigna reticulata*, and *Vigna vexillata* were first considered to carry out preliminary investigations based on the very little information gathered about them and their availability in the nearest gene bank.

Archeological reports suggested that the crop domestication processes were preceded by a period of pre-domestication, where humans first began to purposefully plant wild plants that had favorable traits [11]. This later on led to a purposive and intentional selection of crops with a group of traits, generated through human preferences for ease of harvest and growth advantages under human propagation, known as the domestication syndrome [11]. It is then important to screen the existing traits of the wild crop in order to examine and select crop accessions with traits that humans prefer for an effective domestication. It is from that line of thought that this paper aimed to explore the agro-morphological characteristics of some wild *Vigna* legumes in order to disseminate preliminary information that could lead to their future domestication.

## 2. Materials and Methods

### 2.1. Sample Collection and Preparation

One hundred and sixty (160) accessions of wild *Vigna* species of legumes were obtained from gene banks, as presented in Table S1. Approximately 20–100 seeds of each accession were supplied by the gene banks and planted in pots, which were placed in screen houses at the Nelson Mandela African Institution of Science and Technology, Arusha, Tanzania during the period of November 2017–March 2018. The pot experiment only allowed for seed multiplication and the preliminary observation of the growth behavior of the wild legumes, prior to experimentation in the field. In the field, all the accessions were planted in an experimental plot, following the augmented block design arrangement [12], and allowed to grow until full maturity. In addition to the wild accessions, three domesticated *Vigna* legumes—that is, cowpea (*V. unguiculata*), rice bean (*V. umbellata*), and a semi-domesticated landrace (*V. vexillata*)—were used as checks. The checks were obtained from the Genetic Resource Center (GRC-IITA), Nigeria (cowpea), the National Bureau of Plant Genetic Resources (NBPGR), India (rice bean), and the Australian Grain Gene bank (AGG), Australia (semi-domesticated landrace *V. vexillata*).

### 2.2. Study Sites and Meteorological Considerations

The study was conducted in two agro-ecological zones, located at two research stations in Tanzania, during two main cropping seasons (March–September 2018 and March–September 2019).

The first research station (site A) was at the Tanzania Agricultural Research Institute (TARI), Selian in the Arusha region, located in the northern part of Tanzania. TARI-Selian lies at a latitude of 3°21′50.08″ N and longitude of 36°38′06.29″ E at an elevation of 1390 m above sea level (a.s.l.).

The second site (site B) was at the Tanzania Coffee Research Institute (TaCRI), located in the Hai district, Moshi, Kilimanjaro region (latitude 3°13′59.59″ S, longitude 37°14′54″ E). The site is at an elevation of 1681 m above sea level.

The meteorological characteristics (monthly rainfall and temperature dynamics) of the two study sites for the two study cropping seasons were obtained from the Tanzania Meteorological Agency and are summarized in Figure 1 below.

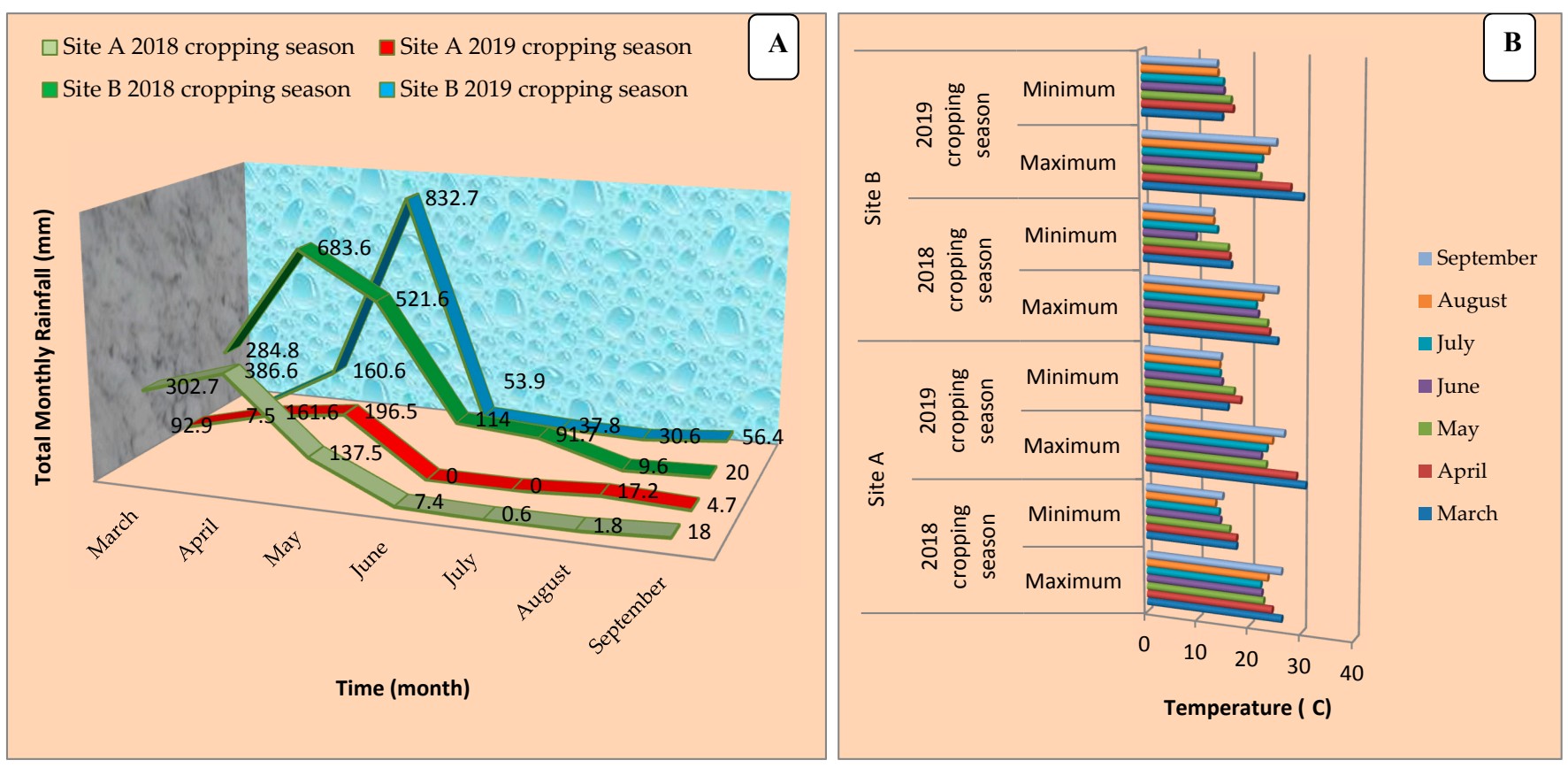

**Figure 1.** Meteorological characteristics of the two study sites. (**A**) rainfall dynamics of the study sites; (**B**) temperature dynamics of the study sites.

## 2.3. Experimental Design and Planting Process

The 160 accessions of wild *Vigna* legumes were planted in an augmented block design field layout, following the randomization generated by the statistical tool on the website developed by the Indian Agricultural Research Institute [13] for 160 accessions, with three checks. The software generated a total of 208 experimental plots, with 8 blocks each containing 26 experimental plots. Each plot represents a line of different wild accession or a check. Each check was repeated two times within a block. Ten seeds from each accession were planted in a line of 5 m in length, with a distance of 50 cm between each seed. The distances between the accessions (lines) within a block, as well as the distance between the blocks, were 1 m each. Data were then collected from five randomly selected plants in each line using the wild *Vigna* descriptors [12].

## 2.4. Data Collection and Analysis

Data on thirty (30) characteristics (both qualitative and quantitative) were recorded using both IPGRI (International Plant Genetic Resource Institute) and NBPGR (National Bureau of Plant Genetic Resources) descriptors [12]. Fifteen (15) qualitative and fifteen (15) quantitative characters were recorded. The descriptors used for the characterization of the wild *Vigna* in this study are found in Table 1 below. Data on the quantitative traits were recorded for five randomly selected individuals per accession.

The generalized linear model procedure (GLM PROC) of the SAS software (SAS University Edition, SAS Institute Inc., North Carolina State, USA) was used to analyze the accession and the block and block vs. accession effects, while the two-way analysis of variance (two-way ANOVA), agglomerative hierarchical clustering (AHC) and principle component analysis (PCA) of XLSTAT were used to analyze the accession × site and accession × season interactions, as well as the clustering and variations among accessions. The SAS University Edition version and the XLSTAT-Base version 21.1.57988.0 were used.

**Table 1.** Important descriptors for the characterization of the wild *Vigna* species germplasm [12].

| | Parameters | Descriptors |
|---|---|---|
| | **Qualitative Traits** | |
| 1 | Seed germination habit | 1. Epigeal, 2. Hypogeal |
| 2 | Attachment of primary leaves (at two-leaf stage) | 1. Sessile, 2. Sub-sessile, 3. Petiolate |
| 3 | Growth habit (recorded at first pod maturity) | 1. Erect, 2. Semi-erect, 3. Spreading, 4. Semi-prostrate, 5. Prostrate, 6. Climbing |
| 4 | Leafiness (at 50% flowering) | 1. Sparse, 2. Intermediate, 3. Abundant |
| 5 | Leaf pubescence | 1. Glabrous, 2. Very sparsely pubescent, 3. Sparsely pubescent, 4. Moderately Pubescent, 5. Densely pubescent |
| 6 | Petiole pubescence | 1. Glabrous, 2. Pubescent, 3. Moderately pubescent, 4. Densely pubescent |
| 7 | Lobing of terminal leaflet (at first pod maturity) | 1. Unlobed, 2. Shallow, 3. Intermediate, 4. Deep, 5. Very deep |
| 8 | Terminal leaflet shape | 1. Lanceolate, 2. Broadly ovate, 3. Ovate, 4. Rhombic, 5. Others |
| 9 | Stipule size | 1. Small, 2. Medium, 3. Large |
| 10 | Hypocotyl color | 1. Green; 2. Purple, 3. Others |
| 11 | Stem pubescence | 1. Glabrous, 2. Sparsely pubescent, 3. Moderately pubescent, 4. Highly pubescent |
| 12 | Pod attachment to peduncle | 1. Erect, 2. Horizontal, 3. Horizontal-pendent 4. Pendent, 5. Others |
| 13 | Pod pubescence | 1. Glabrous, 2. Sparsely pubescent, 3. Moderately pubescent, 4. Densely pubescent |

**Table 1.** *Cont.*

|  | Parameters | Descriptors |
|---|---|---|
| 14 | Pod curvature | 1. Straight, 2. Slightly curved, 3. Curved (sickle shaped) |
| 15 | Constriction of pod between seeds | 1. Absent, 2. Slight, 3. Pronounced |
|  | Pod cross section | 1. Semi flat, 2. Round, 3. Others |
| **Quantitative Traits** | | |
| 1 |  | Germination time (days) |
| 2 |  | Terminal leaflet length (cm) |
| 3 |  | Terminal leaflet width (cm) |
| 4 |  | Petiole length (cm) |
| 5 |  | Days to flowering |
| 6 |  | Flower bud size (cm) |
| 7 |  | Number of flowers per raceme |
| 8 |  | Peduncle length (cm) |
| 9 |  | Pods per peduncle |
| 10 |  | Pod length (cm) |
| 11 |  | Pods per plant |
| 12 |  | Seeds per pod |
| 13 |  | Seed size ($mm^2$) |
| 14 |  | 100-Seed weight (g) |
| 15 |  | Yield (Kg/ha) |

## 3. Results

### 3.1. Qualitative Traits Exploration of the Wild Unexplored Vigna Species

Figure 2 below gives a pictorial description of some distinguishing morphological characteristics of the wild *Vigna* legumes, studied based on the physical phenotypic observations during the pot experimental phase. Other qualitative characteristics were studied in the field at different stages of the plants' growth, i.e., the germination, vegetative, podding, and maturity stages.

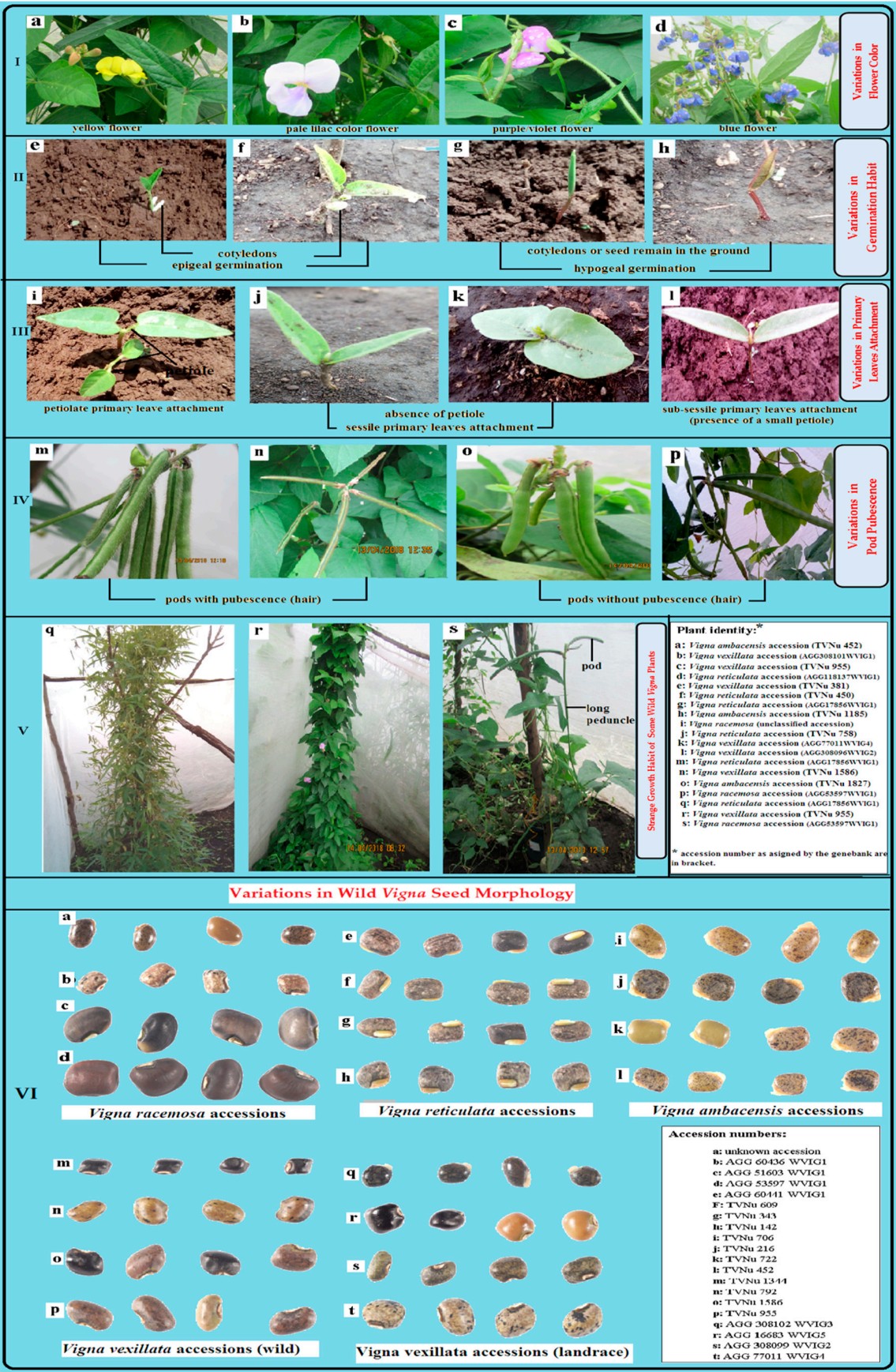

**Figure 2.** Some qualitative morphological characteristics of the studied wild Vigna.

### 3.1.1. Germination Stage

The hypocotyl color, primary leaf attachment (at the two-leaf stage), and germination habit were the traits monitored at this growth stage. Figure 3 shows the percentage of the distribution of accessions for each trait variation. All the checks showed homogenous phenotypic characteristics, while the variations among accessions of the same species were observed for the wild species. *V. ambacensis* accessions showed a higher percentage of purple hypocotyl color, which they share with rice bean (*V. umbellata*). On the other hand, *V. vexillata*, *V. reticulata*, and *V. racemosa* showed higher percentages of green hypocotyl color, similar to cowpea and the landrace of *V. vexillata*.

The *V. vexillata*, *V. ambacensis*, and *V. racemosa* accessions showed a resemblance in the primary leaf attachment trait (sub-sessile) to cowpea, presenting a high percentage of accessions for the trait, while the *V. reticulata* accessions shared the sessile phenotype with the landrace of *V. vexillata*.

Both cowpea and the landrace of *V. vexillata* presented an epigeal germination habit, which they have in common with most accessions of *V. reticulata* and *V. racemosa*, while most accessions of *V. ambacensis* and *V. vexillata* shared a common phenotype (hypogeal) with rice bean.

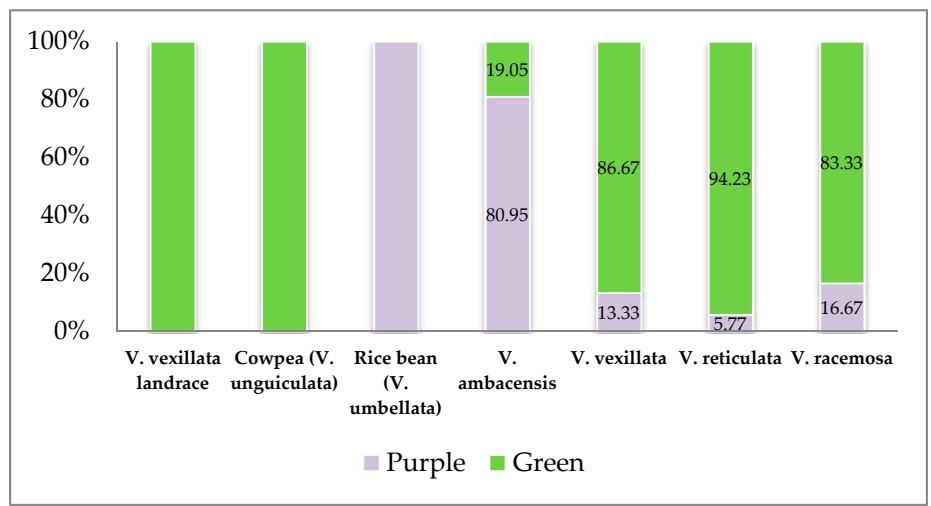

(**a**) Hypocotyl Color.

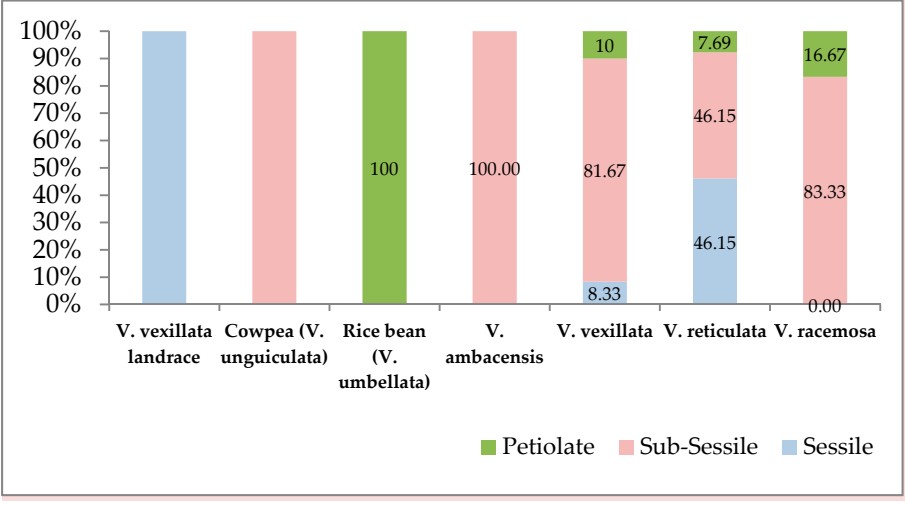

(**b**) Attachment of primary leaves (at the two-leaf stage).

**Figure 3.** *Cont.*

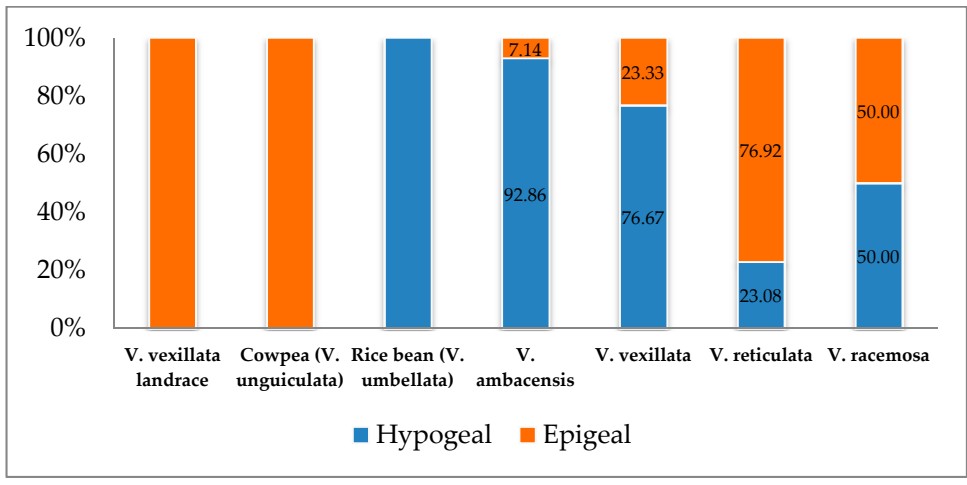

(**c**) Germination Habit.

**Figure 3.** Variations of some selected qualitative traits, evaluated at the germination stage. (**a**) hypocotyl color; (**b**) attachment of primary leaves (at the two-leaf stage); (**c**) germination habit.

### 3.1.2. Vegetative Stage

The frequency distributions of variations for the qualitative traits examined at the vegetative stage are presented in Figure 4. The leafiness, leaf pubescence, petiole pubescence, lobing of terminal leaflet, terminal leaflet shape, stipule size, and stem pubescence traits were monitored to characterize the wild accessions of Vigna legumes at this stage.

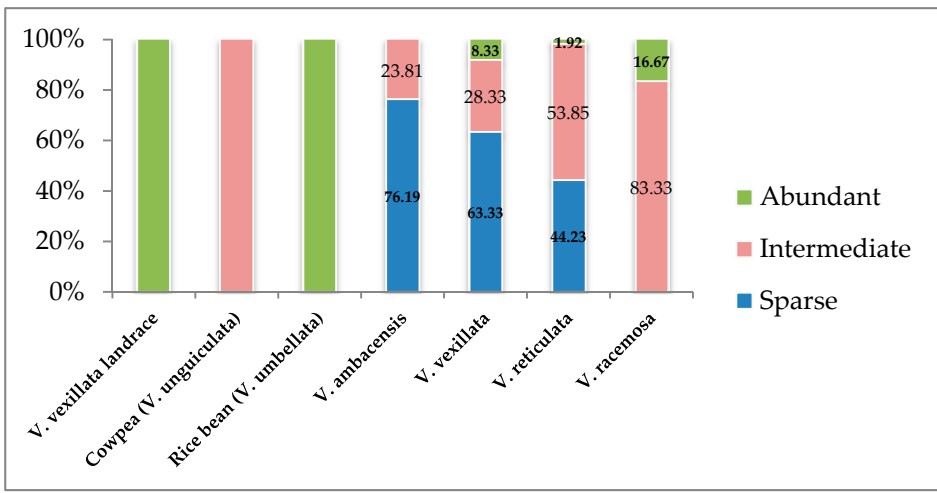

(**a**) Leafiness (at 50% flowering).

**Figure 4.** *Cont*.

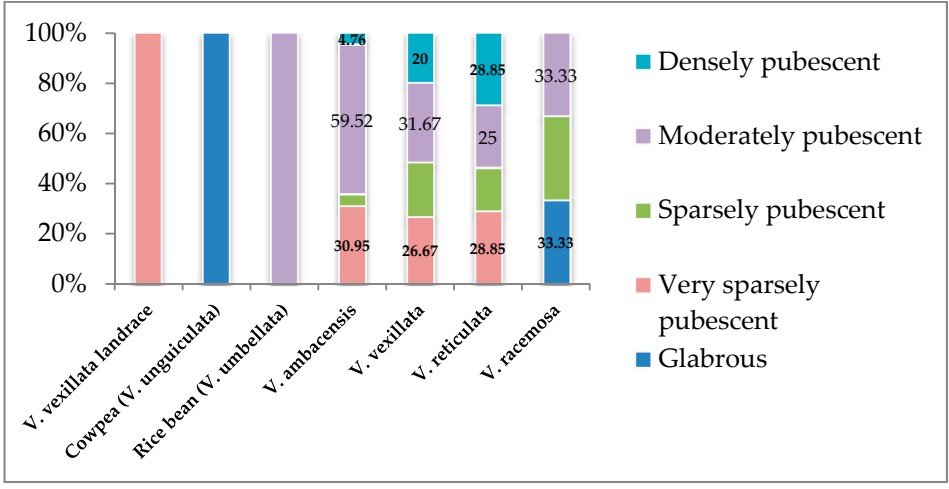

(**b**) Leaf pubescence.

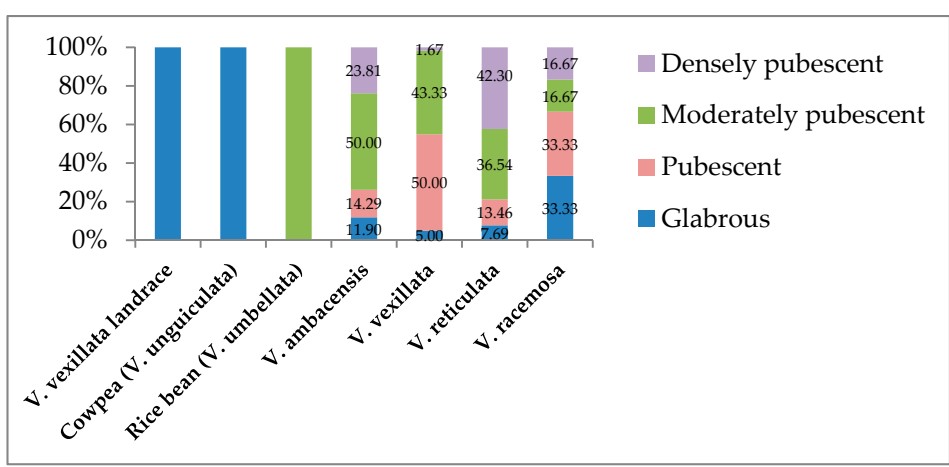

(**c**) Petiole pubescence.

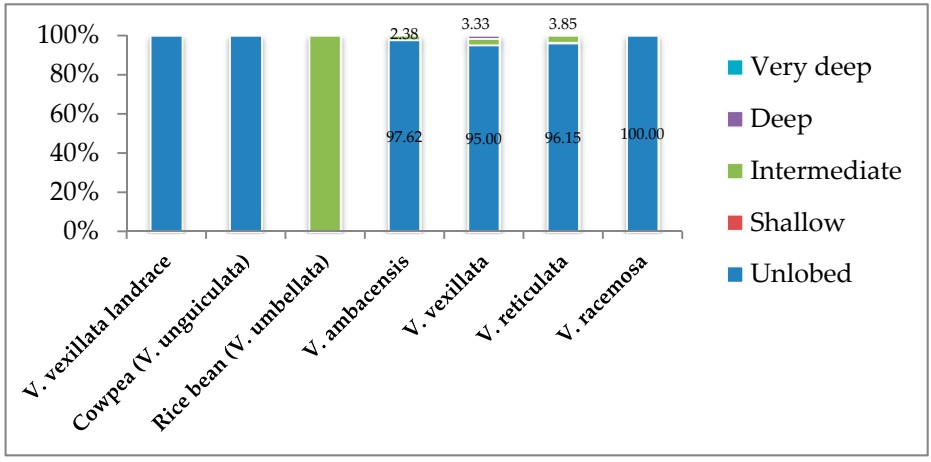

(**d**) Lobing of the terminal leaflet (at first pod maturity).

**Figure 4.** *Cont*.

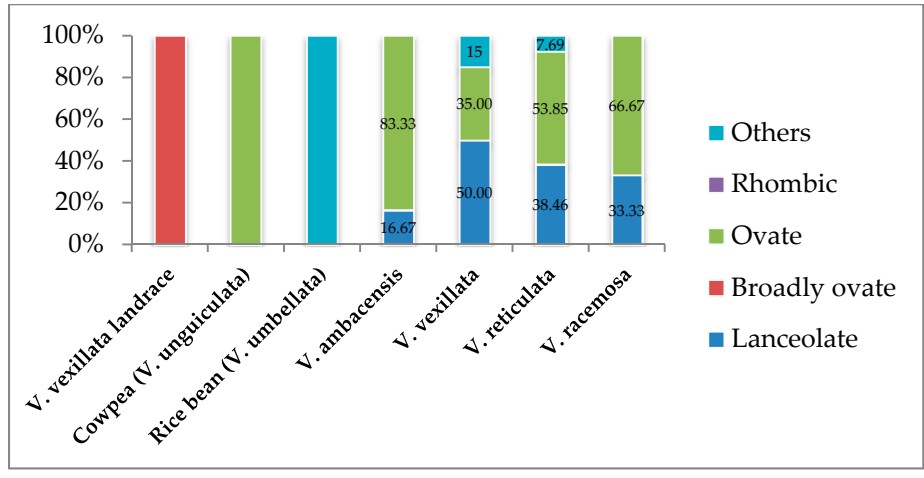

(**e**) Terminal leaflet shape.

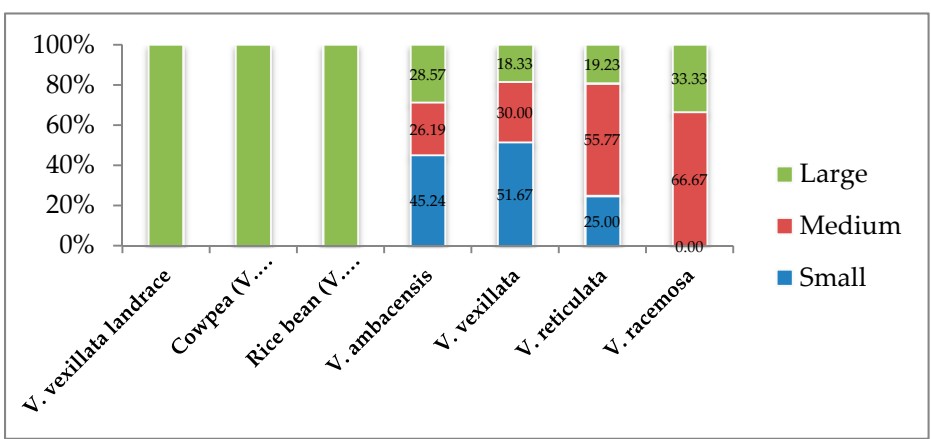

(**f**) Stipule size.

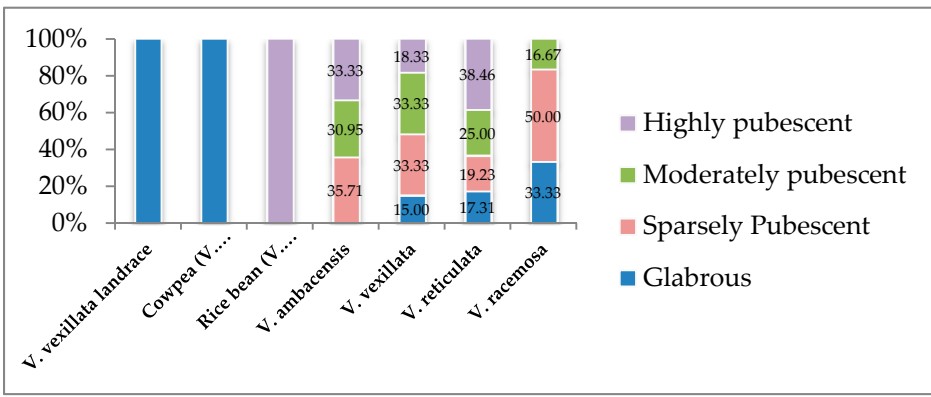

(**g**) Stem pubescence.

**Figure 4.** Variations in some selected qualitative traits, evaluated at the vegetative stage. (**a**) leafiness (at 50% flowering); (**b**) leaf pubescence; (**c**) petiole pubescence; (**d**) lobing of the terminal leaflet (at first pod maturity); (**e**) terminal leaflet shape, (**f**) stipule size; (**g**) stem pubescence.

Most *V. ambacensis* (76%) and *V. vexillata* (63%) accessions presented a sparse leafy character, which was not the case with any of the checks (Figure 4a). Most *V. racemosa* and *V. reticulata* accession shared a common feature of intermediate leafiness with cowpea. Rice bean and the landrace

of *V. vexillata* presented an abundant leafiness, which they had in common with few *V. reticulata*, *V. racemosa*, and *V. vexillata*.

High variations were observed for the leaf pubescence traits among the wild accessions of all the species. A higher percentage (59%) of *V ambacensis* presented moderate leaf pubescence, as found in rice bean, while the other species presented less than 50% accession for the same trait variation. Cowpea had a glabrous leaf pubescence, while the landrace of *V. vexillata* was very sparsely pubescent (Figure 4b).

Considerable variations were also observed in the petiole pubescence trait (Figure 4c). Only the *V. racemosa* accession significantly (33%) shared the common feature of globrous petiole pubescence with cowpea and *V. vexillata* landrace, and 50% of the *V. ambacensis* accession showed a moderately pubescent characteristic, which is similar to that found in rice bean. On the other hand, 50% of the *V. vexillata* accessions were pubescent and did not share the trait intensity with any of the checks. The *V. reticulata* accessions presented the highest percentage (42%) of the densely pubescent variant within the trait and a considerable percentage (36%) of the moderately pubescent variant.

Lobing of the terminal leaflet trait varied little among the studied wild accessions (Figure 4d). The majority (more than 90%) of all the accessions of the studied wild species presented an unlobed variant of the trait, like cowpea and *V. vexillata* landrace (Figure 4d).

Most *V. racemosa*, *V. ambacensis*, and *V. reticulata* (66%, 83% and 53%, respectively) presented an ovate variant of the terminal leaflet shape trait, which is the same variant found in cowpea (Figure 4e). The broadly ovate variant of this trait was only found in the *V. vexillata* landrace, while the rice bean presented an irregular variant (with lobes), which was found in few accessions of *V. reticulata* and *V. vexillata*. The lanceolate variant of the trait was also found in *V. racemosa* (33%), *V. ambacensis* (17%), *V. vexillata* (50%), and *V. reticulata* (38%).

All three of the checks showed a large variant of the stipule size trait, while 67% of *V. racemosa* had the medium stitpule size variant, as well as 56% of the *V. reticulata* accessions (Figure 4f). The small size variant was observed in 45% and 52% of *V. ambacensis* and *V. vexillata*, respectively.

The stem pubescence trait also varied significantly among the wild accessions (Figure 4e). The *V. vexillata* landrace presented the glabrous variant of the trait, which matched with 15.00%, 17.31%, and 33% of *V. vexillata*, *V. reticulata*, and *V. racemosa*, respectively. The stem of rice bean presented the highly pubescent variant of the stem pubescent trait, as found in 33%, 18%, and 38% of *V. ambacensis*, *V. vexillata*, and *V. reticulata*, respectively. A moderately pubescent variant was found in 31%, 33%, 25%, and 17% of *V. ambacensis*, *V. vexillata*, *V. reticulata*, and *V. racemosa*, respectively. Finally, 50% of the *V. racemosa* accessions had the sparsely pubescent variant, while only 36%, 33%, and 19% of *V. ambacensis*, *V. vexillata*, and *V. reticulata* were found to have the same variant.

### 3.1.3. Pod Formation and Maturity Stage

At this stage, the following traits were observed and recorded for wild accessions and checks under study: pod attachment to peducle, pod pubescence, pod curvature, constriction of the pod between seeds, and pod cross section (Figure 5).

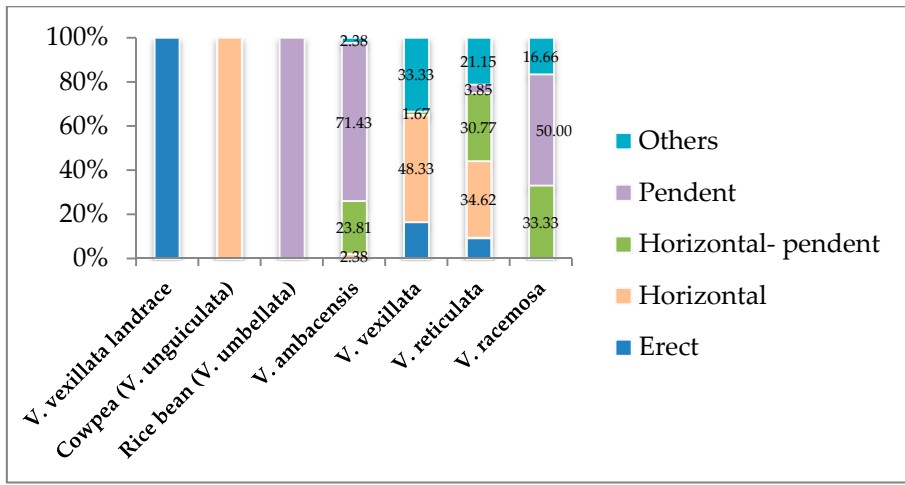

(**a**) Pod attachment to peduncle.

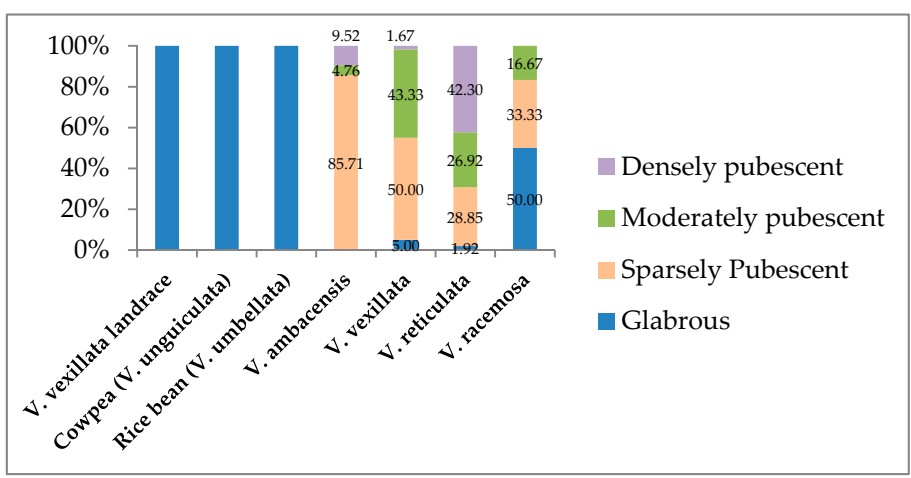

(**b**) Pod pubescence.

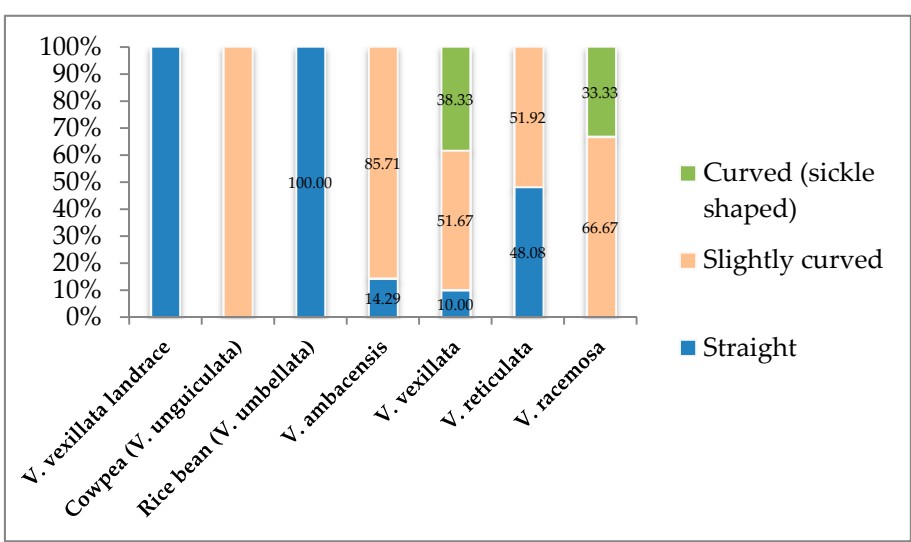

(**c**) Pod curvature.

**Figure 5.** *Cont.*

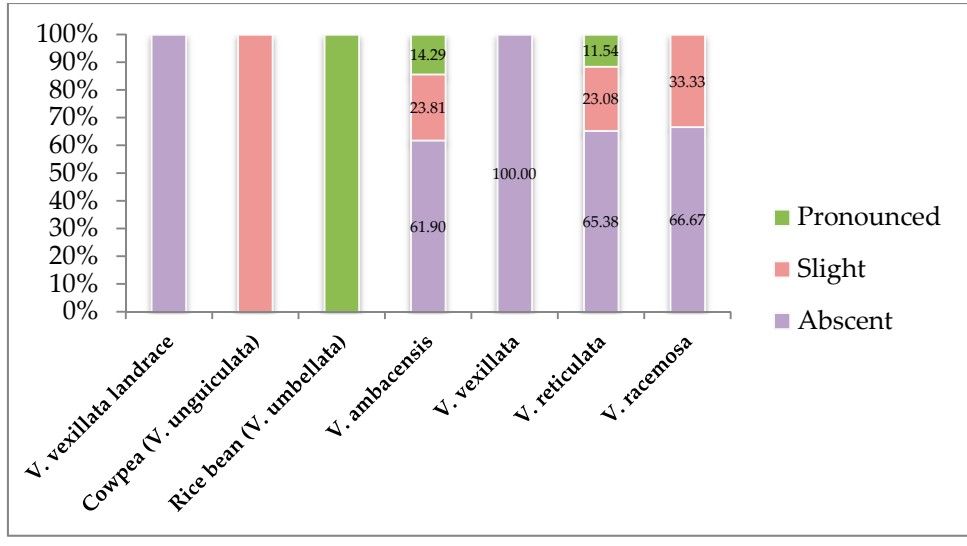

(**d**) Constriction of the pod between seeds.

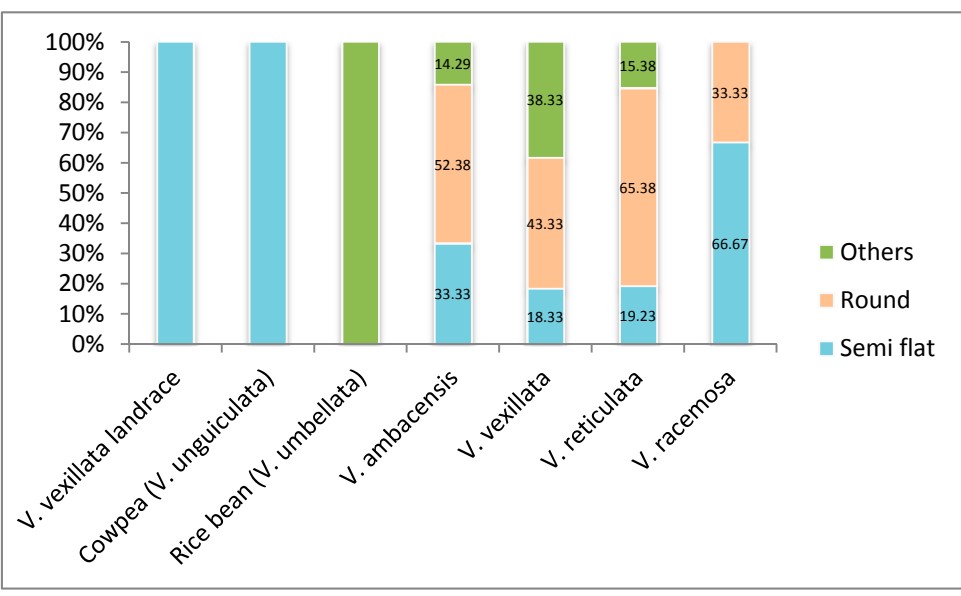

(**e**) Pod cross section.

**Figure 5.** Variations in some selected qualitative traits, evaluated at the maturity and podding stages. (**a**) pod attachment to peduncle; (**b**) pod pubescence; (**c**) pod curvature; (**d**) constriction of the pod between seeds; (**e**) pod cross section.

For the pod attachment to peducle trait, most accessions of *V. ambacensis* (71%) and *V. racemosa* (50%) were similar to rice bean for the ''pendent'' variant of the trait, and 48% of *V. vexillata* and 35% of *V. reticulata* accessions presented the ''horizontal'' variant of the trait, which is similar to cowpea. The ''erect'' variant of the trait was found in *V. vexillata* landrace and a few accessions of *V. vexillata* and *V. reticulata* (Figure 5a). The horizontal-pendent form of the trait was commonly found in all the wild accessions, with 24%, 2%, 31%, and 33% in *V. ambacensis*, *V. vexillata*, *V. reticulata*, and *V. racemosa*, respectively.

All the checks had a glabrous form for the pod pubescence trait (Figure 5b). Only *V. racemosa* (50%) accessions were found to be similar to the checks for this trait. Moreover, 5%, 43%, 27%, and 17% of *V. ambacensis*, *V. vexillata*, *V. reticulata* and *V. racemosa*, respectively, were moderately pubescent, while 86% of *V. ambacensis*, 50% of *V. vexillata*, 29% of *V. reticulata*, and 33% of *V. racemosa* were sparsely

pubescent (Figure 5b). Only 42% of the accessions of *V. reticulata* and 10% of those of the *V. vexillata* pods were densely pubescent.

More than 50% of the studied wild accessions presented the "slightly curved" form of the pod curvature trait, which was similar to cowpea (Figure 5c). Rice bean and the *V. vexillata* landrace commonly shared the "straight" form of the trait with 14% of *V. ambacensis*, 10% of *V. vexillata* and and 48% of *V. reticulata* accessions. On the other hand, only *V. vexillata* (38%) and *V. racemosa* (33%) accessions showed the "curved" form of the pod curvature trait (Figure 5c).

Most of the studied wild Vigna accessions (more than 50%) had no constriction of the pod between seeds (variant: "absent"), as found in *V. vexillata* landrace (Figure 5d). The trait was found in a "pronounced" form only in rice bean and 15% of *V. ambacensis*, as well as 12% of *V. reticulata* accessions. A "slight" constriction of the pod between seeds was the form found in cowpea and 24% of *V. ambacensis*, 23% of *V. reticulata*, and 33% of *V. racemosa* accessions (Figure 5d).

Cowpea and *V. vexillata* landrace presented a "semi-flat" form of the pod cross section trait, together with 33% of *V. ambacensis*, 18% of *V. vexillata*, 19% of *V. reticulata*, and 67% of *V. racemosa* accessions (Figure 5e). Rice bean presented a flat ("other") form of the trait, together with 14% of *V. ambacensis*, 38% of *V. vexillata*, and 15% of *V. reticulata* accessions. The "round" variant of the trait was observed in 52% of *V. ambacensis*, 43% of *V. vexillata*, 65% of *V. reticulata*, and 33% of *V. racemosa* accessions (Figure 5e).

### 3.2. Quantitative Traits Exploration of the Wild Unexplored Vigna Species

Table S2 summarizes the means, ranges, and coefficients of variation for the selected quantitative traits studied at site A and B during the two cropping seasons. Furthermore, the adjusted mean values for the studied traits are summarized per species in Table S3. The two tables show the results for only one season (the 2018 cropping season) for site A (Tables S2 and S3).

To understand the variations of the means for the various traits studied within the cropping sites and seasons, the generalized linear model procedure (glm proc) of the SAS University Editions was run, and the results are summarized in Tables 2–5. One-way analysis of variance (ANOVA) and type III Sum of Squares Analysis, as well as the analysis of differences, helped to indicate the accession effect, block effect, and the differences among the accessions, checks, and check vs. accession.

The results from the site A study during the 2018 cropping season show that there was a significant difference ($p < 0.05$) between the checks and the wild accessions for all the analyzed traits (Table 2a). Accession effects were also found for all the traits, except for the number of flowers per raceme trait (trait 7) (Table 2a). Block effects were only found for the terminal leaflet width (trait 3), days to flowering (trait 5), number of flowers per raceme (trait 7), pods per peduncle (trait 9), pod length (trait 10), and seed size (trait 13) traits (Table 2a). Significant differences among the accessions, checks, and check vs. accession were observed, as shown in Table 2b for all the traits. Exceptionally, the seed size trait showed no significant difference among the checks ($p > 0.05$) (Table 2b).

Similarly, the results from site B during the 2018 cropping season show that there was a significant difference ($p < 0.05$) between the checks and the wild accessions for all the analyzed traits (Table 3a). Accession effects were also found for all the traits, with no exception, like in the case of trait 7 at site A (Table 3a). Block effects were only found for the terminal leaflet width (trait 3), days to flowering (trait 5), number of flowers per raceme (trait 7), pods per peduncle (trait 9), pod length (trait 10), seeds per pod (trait 12), and seed size (trait 13) traits (Table 3a). Significant differences among the accessions, checks, and check vs. accession were observed, as shown in Table 3b for all the traits. Exceptionally, the seed size trait showed no significant difference among the checks ($p > 0.05$) (Table 3b).

A similar pattern of results was observed in the site B study area during the second cropping season (2019 cropping season) (Table 4). It was found that there is a significant difference ($p < 0.05$) between the checks and the wild accessions for all the analyzed traits (Table 4a). Accession effects were also found for all the traits, with no exception, like in the case of trait 7 at site A (Table 4a). Block effects were only found for the terminal leaflet width (trait 3), days to flowering (trait 5), number of

flowers per raceme (trait 7), pods per peduncle (trait 9), pod length (trait 10), seeds per pod (trait 12), and seed size (trait 13) traits (Table 4a). Significant differences among the accessions, checks, and check vs. accession were observed, as shown in Table 4b for all the traits. Exceptionally, the seed size trait showed no significant difference among the checks ($p > 0.05$) (Table 4b).

In order to establish the relationship (interactions) between the accessions and the cropping site and cropping seasons, a two-way analysis of variance was run, and the significant *p*-values are summarized in Table 5. Of the 15 quantitative traits examined, only the days to flowering, pods per plant, hundred seed weight, and the yield were affected by their growing environment (accession x site effect), while only the number of flowers per raceme and the pods per plant were affected by the cropping season (accession x season effect) (Table 5). All the quantitative traits showed significant differences among the accessions for each site and each season. The same result was observed among the checks, except for the seed size trait.

To determine whether some of the wild *Vigna* accessions share common quantitative traits and can be grouped together, an agglomerative hierarchical clustering analysis was performed, and a dendrogram of three clusters was obtained based on 138 accessions out of the 160 planted due to the exclusion of 22 accessions which did not germinate or did not perform well (Figure 6). The various accessions forming each cluster are presented in Table S4. Cluster I, which is made up of the majority of wild accessions also included two checks, the *Vigna vexillata* landrace and cowpea (*V. unguiculata*). Cluster II was made up of only check 3 (rice bean, *V. umbellate*), while cluster III contained 50 accessions of the wild *Vigna* species.

Furthermore, to examine the relationship that could exist between the quantitative traits and the accessions, as well as the relationship between the accessions themselves, a principal component analysis (XLSTAT) was performed using the adjusted means values, obtained earlier. A correlation circle, combined with an observation chart, was obtained, as shown in Figure 7. The analysis showed that the first (F1 = 45.39%) and second (F2 = 14.22%) PCA dimensions represent 59.61% of the initial information, which is the best combination and explains the variation among the accessions and traits. It was found that there is a positive correlation between the traits, except for the ''days to flowering'' trait, which is due to the angles between their vectors (Figure 7). It was also noted that all the checks, together with a set of wild accessions, are found on the right side of the F1 axis, forming a group of accessions with higher values for the examined quantitative traits, except for the days to flowering trait. Those accessions could share common features with the checks. A second group, made up of only wild accessions, was found on the left side of the F1 axis, representing the accessions with lower values for the evaluated traits. These accessions also present lower values for the ''days to flowering'' trait on the F2 axis (Figure 7).

**Table 2.** (**a**) Analysis of variance (ANOVA) and type III sum of squares analysis for the selected quantitative traits* at the Tanzania Agricultural Research Institute (TARI, Arusha region) during the 2018 cropping season; (**b**) analysis of the differences and interactions between accessions and checks for the selected traits (all species) at the Tanzania Agricultural Research Institute (TARI, Arusha region) during the 2018 cropping season.

**(a)**

| | | ANOVA | | | | | Type III Sum of Squares Analysis | | | | | | | | |
| | | Model | | | | | Block Effect | | | | | Accession Effect | | | |
| Traits | DF | Sum of Squares | Mean Squares | F | Pr > F | DF | Sum of Squares | Mean Squares | F | p | DF | Sum of Squares | Mean Squares | F | p |
|---|---|---|---|---|---|---|---|---|---|---|---|---|---|---|---|
| 1 | 172 | 230.50 | 1.34 | 0.00 | <0.0001 | 7 | 0.00 | 0.00 | - | - | 165 | 230.19 | 1.40 | 0.00 | <0.0001 |
| 2 | 172 | 2513.26 | 14.61 | 11.86 | <0.0001 | 7 | 36.22 | 5.17 | 4.20 | 0.0019 | 165 | 1512.31 | 9.17 | 7.44 | <0.0001 |
| 3 | 172 | 685.22 | 3.98 | 2.99 | 0.0001 | 7 | 15.18 | 2.17 | 1.63 | 0.1600 | 165 | 675.48 | 4.09 | 3.07 | 0.0001 |
| 4 | 172 | 1999.94 | 11.63 | 6.57 | <0.0001 | 7 | 72.68 | 10.38 | 5.86 | 0.0001 | 165 | 1908.16 | 11.56 | 6.53 | <0.0001 |
| 5 | 172 | 41,888.19 | 243.54 | 269.88 | <0.0001 | 7 | 2.67 | 0.38 | 0.42 | 0.8818 | 165 | 41,253.80 | 250.02 | 277.07 | <0.0001 |
| 6 | 172 | 92.42 | 0.54 | 21.64 | <0.0001 | 7 | 0.63 | 0.09 | 3.65 | 0.0046 | 165 | 75.72 | 0.46 | 18.48 | <0.0001 |
| 7 | 172 | 5907.56 | 34.35 | 14.23 | <0.0001 | 7 | 24.98 | 3.57 | 1.48 | 0.2069 | 165 | 165 | 4000.12 | 24.24 | 10.05 |
| 8 | 172 | 16,033.67 | 93.22 | 68.36 | <0.0001 | 7 | 25.65 | 3.66 | 2.69 | 0.0245 | 165 | 12,837.61 | 77.80 | 57.05 | <0.0001 |
| 9 | 172 | 759.23 | 4.41 | 4.96 | <0.0001 | 7 | 10.55 | 1.51 | 1.69 | 0.1427 | 165 | 539.13 | 3.27 | 3.67 | <0.0001 |
| 10 | 172 | 4990.46 | 29.01 | 27.94 | <0.0001 | 7 | 13.24 | 1.89 | 1.82 | 0.1139 | 165 | 3894.89 | 23.61 | 22.73 | <0.0001 |
| 11 | 172 | 298,644.75 | 1736.31 | 365.40 | <0.0001 | 7 | 81.81 | 11.69 | 2.46 | 0.0367 | 165 | 221,943.38 | 1345.11 | 283.07 | <0.0001 |
| 12 | 172 | 3427.67 | 19.93 | Infini | <0.0001 | 7 | 0.00 | 0.00 | | | 165 | 2597.62 | 15.74 | Infini | <0.0001 |
| 13 | 172 | 26,079.22 | 151.62 | 29.99 | <0.0001 | 7 | 49.43 | 7.06 | 1.40 | 0.2377 | 165 | 23,910.77 | 144.91 | 28.66 | <0.0001 |
| 14 | 172 | 6155.01 | 35.78 | 14.06 | <0.0001 | 7 | 60.79 | 8.68 | 3.41 | 0.0070 | 165 | 5923.05 | 35.90 | 14.11 | <0.0001 |
| 15 | 172 | 225,200,114.2 | 1,309,303.0 | 14.99 | <0.0001 | 7 | 2,007,022.1 | 286,717.4 | 3.28 | 0.0087 | 165 | 218,001,052.5 | 1,321,218.5 | 15.13 | <0.0001 |

**(b)**

| | | Contrast (Differences) | | | | | | | | | | | | | |
| | | Among Accessions | | | | | Among Checks | | | | | Check vs. Accession | | | |
| Traits | DF | Sum of Squares | Mean Squares | F | Pr > F | DF | Sum of Squares | Mean Squares | F | p | DF | Sum of Squares | Mean Squares | F | p |
|---|---|---|---|---|---|---|---|---|---|---|---|---|---|---|---|
| 1 | 53 | 36.59 | 0.69 | Infini | <0.0001 | 3 | 16.00 | 5.33 | Infini | <0.0001 | 1 | 49.51 | 49.51 | Infini | <0.0001 |
| 2 | 53 | 397.53 | 7.50 | 6.09 | <0.0001 | 3 | 80.92 | 26.97 | 21.89 | <0.0001 | 1 | 49.72 | 49.72 | 40.35 | <0.0001 |
| 3 | 53 | 181.78 | 3.43 | 2.58 | 0.0019 | 3 | 28.15 | 9.38 | 7.05 | 0.0008 | 1 | 130.73 | 130.73 | 98.16 | <0.0001 |
| 4 | 53 | 779.54 | 14.71 | 8.31 | <0.0001 | 3 | 132.98 | 44.33 | 25.03 | <0.0001 | 1 | 343.41 | 343.41 | 193.94 | <0.0001 |
| 5 | 53 | 17,494.29 | 330.08 | 365.79 | <0.0001 | 3 | 55.09 | 18.36 | 20.35 | <0.0001 | 1 | 1095.79 | 1095.79 | 1214.34 | <0.0001 |
| 6 | 53 | 17.56 | 0.33 | 13.34 | <0.0001 | 3 | 0.29 | 0.10 | 3.84 | 0.0178 | 1 | 0.29 | 0.29 | 11.70 | 0.0016 |
| 7 | 53 | 984.25 | 18.57 | 7.70 | <0.0001 | 3 | 137.32 | 45.77 | 18.97 | <0.0001 | 1 | 159.39 | 159.39 | 66.05 | <0.0001 |

**Table 2.** *Cont.*

| | | | | | | | | | | | | | | | |
|---|---|---|---|---|---|---|---|---|---|---|---|---|---|---|---|
| 8 | 53 | 1790.74 | 33.79 | 24.78 | <0.0001 | 3 | 240.59 | 80.20 | 58.81 | <0.0001 | 1 | 1751.07 | 1751.07 | 1284.07 | <0.0001 |
| 9 | 53 | 141.29 | 2.67 | 3.00 | 0.0004 | 3 | 27.70 | 9.23 | 10.38 | <0.0001 | 1 | 10.15 | 10.15 | 11.41 | 0.0018 |
| 10 | 53 | 822.94 | 15.53 | 14.95 | <0.0001 | 3 | 516.15 | 172.049 | 165.66 | <0.0001 | 1 | 91.85 | 91.85 | 88.45 | <0.0001 |
| 11 | 53 | 98,712.64 | 1862.50 | 391.96 | <0.0001 | 3 | 1092.09 | 364.03 | 76.61 | <0.0001 | 1 | 6392.80 | 6392.80 | 1345.35 | <0.0001 |
| 12 | 53 | 377.84 | 7.13 | Infini | <0.0001 | 3 | 288.00 | 96.00 | Infini | <0.0001 | 1 | 82.51 | 82.51 | Infini | <0.0001 |
| 13 | 53 | 8853.77 | 167.05 | 33.04 | <0.0001 | 3 | 5.64 | 1.88 | 0.37 | 0.7736 | 1 | 1092.99 | 1092.99 | 216.20 | <0.0001 |
| 14 | 53 | 2595.73 | 48.98 | 19.24 | <0.0001 | 3 | 59.56 | 19.85 | 7.80 | 0.0004 | 1 | 379.86 | 379.86 | 149.26 | <0.0001 |
| 15 | 53 | 103,443,899.9 | 1,951,771.7 | 22.35 | <0.0001 | 3 | 13,494,835.8 | 4,498,278.6 | 51.52 | <0.0001 | 1 | 5,333,366.9 | 5,333,366.9 | 61.08 | <0.0001 |

\* **1:** Germination time; **2:** Terminal leaflet length; **3:** Terminal leaflet width; **4:** Petiole length; **5:** Days to flowering; **6:** Flower bud size; **7:** Number of flowers per raceme; **8:** Peduncle length; **9:** Pods per peduncle; **10:** Pod length; **11:** Pods per plant; **12:** Seeds per pod; **13:** Seed size; **14:** 100-Seed weight; **15:** Yield.

**Table 3.** (**a**) Analysis of variance (ANOVA) and type III sum of squares analysis for the selected quantitative traits\* at the Tanzania Coffee Research Institute (TaCRI, Kilimanjaro region) during the 2018 cropping season; (**b**) analysis of the differences and interactions between accessions and checks for the selected traits (all species) at the Tanzania Coffee Research Institute (TaCRI, Kilimanjaro region) during the 2018 cropping season.

| (a) | | | | | | | | | | | | | | | |
|---|---|---|---|---|---|---|---|---|---|---|---|---|---|---|---|
| | | **ANOVA** | | | | | **Type III Sum of Square Analysis** | | | | | | | | |
| **Traits** | | **Model** | | | | | **Block Effect** | | | | | **Accession Effect** | | | |
| | **DF** | **Sum of Squares** | **Mean Squares** | **F** | **Pr > F** | **DF** | **Sum of Squares** | **Mean Squares** | **F** | ***p*** | **DF** | **Sum of Squares** | **Mean Squares** | **F** | ***p*** |
| 1 | 172 | 2564.76 | 14.91 | Infini | <0.0001 | 7 | 0.00 | 0.00 | | | 165 | 1217.73 | 7.38 | Infini | <0.0001 |
| 2 | 172 | 2986.01 | 17.36 | 11.86 | <0.0001 | 7 | 43.03 | 6.15 | 4.20 | 0.0019 | 165 | 1796.78 | 10.89 | 7.44 | <0.0001 |
| 3 | 172 | 829.12 | 4.82 | 2.99 | 0.0001 | 7 | 18.36 | 2.62 | 1.63 | 0.1600 | 165 | 817.33 | 4.95 | 3.07 | 0.0001 |
| 4 | 172 | 2121.73 | 12.33 | 6.57 | <0.0001 | 7 | 77.10 | 11.01 | 5.86 | 0.0001 | 165 | 2024.37 | 12.27 | 6.53 | <0.0001 |
| 5 | 172 | 45,758.01 | 266.03 | 233.27 | <0.0001 | 7 | 4.33 | 0.62 | 0.54 | 0.80 | 165 | 44,869.16 | 271.93 | 238.44 | <0.0001 |
| 6 | 172 | 133.09 | 0.77 | 21.64 | <0.0001 | 7 | 0.91 | 0.13 | 3.65 | 0.0046 | 165 | 109.03 | 0.66 | 18.48 | <0.0001 |
| 7 | 172 | 4330.26 | 25.18 | 8.55 | <0.0001 | 7 | 22.38 | 3.20 | 1.09 | 0.3929 | 165 | 3090.50 | 18.73 | 6.36 | <0.0001 |
| 8 | 172 | 17,010.12 | 98.90 | 68.36 | <0.0001 | 7 | 27.21 | 3.89 | 2.69 | 0.0245 | 165 | 13,619.42 | 82.54 | 57.05 | <0.0001 |
| 9 | 172 | 748.83 | 4.35 | 4.82 | <0.0001 | 7 | 10.10 | 1.57 | 1.74 | 0.1314 | 165 | 538.18 | 3.26 | 3.61 | <0.0001 |
| 10 | 172 | 4792.84 | 27.87 | 27.94 | <0.0001 | 7 | 12.72 | 1.817 | 1.82 | 0.1139 | 165 | 3740.65 | 22.67 | 22.73 | <0.0001 |
| 11 | 172 | 60,475.56 | 351.60 | 365.40 | <0.0001 | 7 | 16.57 | 2.37 | 2.46 | 0.0367 | 165 | 44,943.53 | 272.39 | 283.07 | <0.0001 |
| 12 | 172 | 3387.31 | 19.69 | 236.32 | <0.0001 | 7 | 0.58 | 0.08 | 1.00 | 0.4478 | 165 | 2581.24 | 15.64 | 187.73 | <0.0001 |
| 13 | 172 | 23,536.50 | 136.84 | 29.99 | <0.0001 | 7 | 44.61 | 6.37 | 1.40 | 0.2377 | 165 | 21,579.47 | 130.78 | 28.66 | <0.0001 |
| 14 | 172 | 4993.97 | 29.03 | 14.40 | <0.0001 | 7 | 48.15 | 6.88 | 3.41 | 0.0070 | 165 | 4836.06 | 29.31 | 14.54 | <0.0001 |
| 15 | 172 | 182,274,678.9 | 1,059,736.5 | 15.32 | <0.0001 | 7 | 1,589,762.2 | 227,108.9 | 3.28 | 0.0087 | 165 | 177,020,077.70 | 1,072,849.00 | 15.51 | <0.0001 |

**Table 3.** *Cont.*

| | | | | | | | | | | | | | | | |
|---|---|---|---|---|---|---|---|---|---|---|---|---|---|---|---|
| | | | | | | (b) | | | | | | | | | |
| | | | | | | Contrast (Differences) | | | | | | | | | |
| Traits | | Among Accessions | | | | | Among Checks | | | | | Check vs. Accession | | | |
| | DF | Sum of Squares | Mean Squares | F | Pr > F | DF | Sum of Squares | Mean Squares | F | *p* | DF | Sum of Squares | Mean Squares | F | *p* |
| 1 | 53 | 67.20 | 1.27 | Infini | <0.0001 | 3 | 16.00 | 5.33 | Infini | <0.0001 | 1 | 118.26 | 118.26 | Infini | <0.0001 |
| 2 | 53 | 472.30 | 8.91 | 6.09 | <0.0001 | 3 | 96.14 | 32.04 | 21.89 | <0.0001 | 1 | 59.07 | 59.07 | 40.35 | <0.0001 |
| 3 | 53 | 219.95 | 4.15 | 2.58 | 0.0019 | 3 | 34.07 | 11.36 | 7.05 | 0.0008 | 1 | 158.18 | 158.18 | 98.16 | <0.0001 |
| 4 | 53 | 827.02 | 15.60 | 8.31 | <0.0001 | 3 | 141.08 | 47.03 | 25.03 | <0.0001 | 1 | 364.33 | 364.33 | 193.94 | <0.0001 |
| 5 | 53 | 17,480.49 | 329.82 | 289.20 | <0.0001 | 3 | 58.84375 | 19.61 | 17.20 | <0.0001 | 1 | 872.40 | 872.40 | 764.95 | <0.0001 |
| 6 | 53 | 25.28 | 0.47 | 13.34 | <0.0001 | 3 | 0.41 | 0.14 | 3.84 | 0.0178 | 1 | 0.42 | 0.42 | 11.70 | 0.0016 |
| 7 | 53 | 805.51 | 15.20 | 5.16 | <0.0001 | 3 | 125.78 | 41.93 | 14.25 | <0.0001 | 1 | 169.17 | 169.17 | 57.48 | <0.0001 |
| 8 | 53 | 1899.80 | 35.84 | 24.78 | <0.0001 | 3 | 255.25 | 85.08 | 58.81 | <0.0001 | 1 | 1857.71 | 1857.71 | 1284.07 | <0.0001 |
| 9 | 53 | 138.13 | 2.61 | 2.89 | 0.0006 | 3 | 29.55375 | 9.85 | 10.92 | <0.0001 | 1 | 12.05 | 12.05 | 13.35 | 0.0008 |
| 10 | 53 | 790.35 | 14.91 | 14.95 | <0.0001 | 3 | 495.71 | 165.24 | 165.66 | <0.0001 | 1 | 88.22 | 88.22 | 88.45 | <0.0001 |
| 11 | 53 | 19,989.31 | 377.16 | 391.96 | <0.0001 | 3 | 221.15 | 73.72 | 76.61 | <0.0001 | 1 | 1294.54 | 1294.54 | 1345.35 | <0.0001 |
| 12 | 53 | 377.20 | 7.12 | 85.40 | <0.0001 | 3 | 276.38 | 92.13 | 1105.50 | <0.0001 | 1 | 78.07 | 78.070 | 936.85 | <0.0001 |
| 13 | 53 | 7990.53 | 150.76 | 33.04 | <0.0001 | 3 | 5.093673 | 1.70 | 0.37 | 0.7736 | 1 | 986.42 | 986.43 | 216.20 | <0.0001 |
| 14 | 53 | 2134.87 | 40.28 | 19.98 | <0.0001 | 3 | 47.18 | 15.73 | 7.80 | 0.0004 | 1 | 324.85 | 324.85 | 161.15 | <0.0001 |
| 15 | 53 | 83,831,969.77 | 1,581,735.28 | 22.87 | <0.0001 | 3 | 10,689,259.48 | 3,563,086.49 | 51.52 | <0.0001 | 1 | 4,556,054.50 | 4,556,054.50 | 65.87 | <0.0001 |

\* **1**: Germination time; **2**: Terminal leaflet length; **3**: Terminal leaflet width; **4**: Petiole length; **5**: Days to flowering; **6**: Flower bud size; **7**: Number of flowers per raceme; **8**: Peduncle length; **9**: Pods per peduncle; **10**: Pod length; **11**: Pods per plant; **12**: Seeds per pod; **13**: Seed size; **14**: 100-Seed weight; **15**: Yield.

**Table 4.** (**a**) Analysis of variance (ANOVA) and type III sum of squares analysis for the selected quantitative traits \* at the Tanzania Coffee Research Institute (TaCRI, Kilimanjaro region) during the 2019 cropping season; (**b**) analysis of the differences and interactions between accessions and checks for the selected traits (all species) at the Tanzania Coffee Research Institute (TaCRI, Kilimanjaro region) during the 2019 cropping season.

| | | | | | | | (a) | | | | | | | | |
|---|---|---|---|---|---|---|---|---|---|---|---|---|---|---|---|
| | | ANOVA | | | | | | | | Type III Sum of Square Analysis | | | | | |
| Traits | | Model | | | | | Block Effect | | | | | AccessionEffect | | | |
| | DF | Sum of Squares | Mean Squares | F | Pr > F | DF | Sum of Squares | Mean Squares | F | *p* | DF | Sum of Squares | Mean Squares | F | *p* |
| 1 | 172 | 1886.23 | 10.97 | Infini | <0.0001 | 7 | 0.00 | 0.00 | | | 165 | 1236.69 | 7.50 | Infini | <0.0001 |
| 2 | 172 | 2312.36 | 13.44 | 11.86 | <0.0001 | 7 | 33.32 | 4.76 | 4.20 | 0.0019 | 165 | 1391.43 | 8.43 | 7.44 | <0.0001 |

**Table 4.** *Cont.*

| | DF | Sum | Mean | F | Pr>F | DF | Sum | Mean | F | p | DF | Sum | Mean | F | p |
|---|---|---|---|---|---|---|---|---|---|---|---|---|---|---|---|
| 3 | 172 | 627.56 | 3.65 | 2.99 | 0.0001 | 7 | 13.90 | 1.99 | 1.63 | 0.1600 | 165 | 618.64 | 3.75 | 3.07 | 0.0001 |
| 4 | 172 | 2185.86 | 12.71 | 6.57 | <0.0001 | 7 | 79.43 | 11.35 | 5.86 | 0.0001 | 165 | 2085.55 | 12.64 | 6.53 | <0.0001 |
| 5 | 172 | 45,758.01 | 266.03 | 233.27 | <0.0001 | 7 | 4.33 | 0.62 | 0.54 | 0.7960 | 165 | 44,869.16 | 271.93 | 238.44 | <0.0001 |
| 6 | 172 | 155.23 | 0.90 | 21.64 | <0.0001 | 7 | 1.07 | 0.15 | 3.65 | 0.0046 | 165 | 127.18 | 0.77 | 18.48 | <0.0001 |
| 7 | 172 | 108,256.43 | 629.40 | 8.55 | <0.0001 | 7 | 559.500 | 79.93 | 1.09 | 0.3929 | 165 | 77,262.23 | 468.26 | 6.36 | <0.0001 |
| 8 | 172 | 18,046.03 | 104.92 | 68.36 | <0.0001 | 7 | 28.86 | 4.12 | 2.69 | 0.0245 | 165 | 14,448.84 | 87.57 | 57.05 | <0.0001 |
| 9 | 172 | 1643.58 | 9.56 | 4.62 | <0.0001 | 7 | 26.73 | 3.82 | 1.84 | 0.1094 | 165 | 1181.19 | 7.16 | 3.46 | <0.0001 |
| 10 | 172 | 4918.26 | 28.59 | 27.94 | <0.0001 | 7 | 13.05 | 1.86 | 1.82 | 0.1139 | 165 | 3838.54 | 23.26 | 22.73 | <0.0001 |
| 11 | 172 | 216,024.75 | 1255.96 | 365.40 | <0.0001 | 7 | 59.18 | 8.45 | 2.46 | 0.0367 | 165 | 160,542.80 | 972.99 | 283.07 | <0.0001 |
| 12 | 172 | 4479.72 | 26.04 | 236.32 | <0.0001 | 7 | 0.77 | 0.11 | 1.00 | 0.4478 | 165 | 3413.69 | 20.69 | 187.73 | <0.0001 |
| 13 | 172 | 24,009.58 | 139.59 | 29.99 | <0.0001 | 7 | 45.51 | 6.50 | 1.40 | 0.2377 | 165 | 22,013.22 | 133.41 | 28.66 | <0.0001 |
| 14 | 172 | 5554.90 | 32.30 | 14.06 | <0.0001 | 7 | 54.86 | 7.84 | 3.41 | 0.0070 | 165 | 5345.55 | 32.40 | 14.11 | <0.0001 |
| 15 | 172 | 203,243,103.1 | 1,181,645.9 | 14.99 | <0.0001 | 7 | 1,811,337.5 | 258,762.5 | 3.28 | 0.0087 | 165 | 196,745,949.9 | 1,192,399.7 | 15.13 | <0.0001 |

(b)

| | Contrast (Differences) | | | | | | | | | | | | | | |
|---|---|---|---|---|---|---|---|---|---|---|---|---|---|---|---|
| Traits | Among Accessions | | | | | Among Checks | | | | | Check vs. Accession | | | | |
| | DF | Sum of Squares | Mean Squares | F | Pr > F | DF | Sum of Squares | Mean Squares | F | p | DF | Sum of Squares | Mean Squares | F | p |
| 1 | 53 | 44.13 | 0.83 | Infini | <0.0001 | 3 | 16.00 | 5.33 | Infini | <0.0001 | 1 | 103.45 | 103.45 | Infini | <0.0001 |
| 2 | 53 | 365.75 | 6.90 | 6.09 | <0.0001 | 3 | 74.45 | 24.82 | 21.89 | <0.0001 | 1 | 45.74 | 45.74 | 40.35 | <0.0001 |
| 3 | 53 | 166.48 | 3.14 | 2.58 | 0.0019 | 3 | 25.78 | 8.59 | 7.05 | 0.0008 | 1 | 119.73 | 119.73 | 98.16 | <0.0001 |
| 4 | 53 | 852.01 | 16.08 | 8.31 | <0.0001 | 3 | 145.34 | 48.45 | 25.03 | <0.0001 | 1 | 375.34 | 375.34 | 193.94 | <0.0001 |
| 5 | 53 | 17,480.49 | 329.82 | 289.20 | <0.0001 | 3 | 58.84 | 19.61 | 17.20 | <0.0001 | 1 | 872.40 | 872.40 | 764.95 | <0.0001 |
| 6 | 53 | 29.49 | 0.56 | 13.34 | <0.0001 | 3 | 0.48 | 0.16 | 3.84 | 0.0178 | 1 | 0.49 | 0.49 | 11.70 | 0.0016 |
| 7 | 53 | 20,137.78 | 379.96 | 5.16 | <0.0001 | 3 | 3144.59 | 1048.20 | 14.25 | <0.0001 | 1 | 4229.18 | 4229.18 | 57.48 | <0.0001 |
| 8 | 53 | 2015.50 | 38.03 | 24.78 | <0.0001 | 3 | 270.79 | 90.26 | 58.81 | <0.0001 | 1 | 1970.84 | 1970.84 | 1284.07 | <0.0001 |
| 9 | 53 | 295.70 | 5.58 | 2.70 | 0.0013 | 3 | 63.23 | 21.08 | 10.18 | <0.0001 | 1 | 25.27 | 25.27 | 12.21 | 0.0013 |
| 10 | 53 | 811.04 | 15.30 | 14.95 | <0.0001 | 3 | 508.68 | 169.56 | 165.66 | <0.0001 | 1 | 90.53 | 90.53 | 88.45 | <0.0001 |
| 11 | 53 | 71,403.81 | 1347.24 | 391.96 | <0.0001 | 3 | 789.97 | 263.32 | 76.61 | <0.0001 | 1 | 4624.23 | 4624.23 | 1345.35 | <0.0001 |
| 12 | 53 | 498.84 | 9.41 | 85.40 | <0.0001 | 3 | 365.51 | 121.84 | 1105.50 | <0.0001 | 1 | 103.25 | 103.25 | 936.85 | <0.0001 |
| 13 | 53 | 8151.14 | 153.80 | 33.04 | <0.0001 | 3 | 5.20 | 1.70 | 0.37 | 0.7736 | 1 | 1006.25 | 1006.25 | 216.20 | <0.0001 |
| 14 | 53 | 2342.65 | 44.20 | 19.24 | <0.0001 | 3 | 53.76 | 17.92 | 7.80 | 0.0004 | 1 | 342.83 | 342.83 | 149.26 | <0.0001 |
| 15 | 53 | 93,358,119.63 | 1,761,473.96 | 22.35 | <0.0001 | 3 | 12,179,089.35 | 4,059,696.45 | 51.52 | <0.0001 | 1 | 4,813,363.60 | 4,813,363.60 | 61.08 | <0.0001 |

* **1:** Germination time; **2:** Terminal leaflet length; **3:** Terminal leaflet width; **4:** Petiole length; **5:** Days to flowering; **6:** Flower bud size; **7:** Number of flowers per raceme; **8:** Peduncle length; **9:** Pods per peduncle; **10:** Pod length; **11:** Pods per plant; **12:** Seeds per pod; **13:** Seed size; **14:** 100-Seed weight; **15:** Yield.

**Table 5.** Two-way analysis of variance for the interactions due to the site and season for the studied quantitative traits.

| S/N | Traits | *p*-Values for Site Effects | | | *p*-Values for Season Effects | | |
|---|---|---|---|---|---|---|---|
| | | Site | Accession | Site x Accession | Season | Accession | Accession x Season |
| 1 | Germination time (days) | <0.0001 | 0.000 | 0.153 | 0.097 | 0.000 | 0.979 |
| 2 | Terminal leaflet length (cm) | <0.0001 | <0.0001 | 1.000 | <0.0001 | <0.0001 | 0.961 |
| 3 | Terminal leaflet width (cm) | 0.000 | <0.0001 | 1.000 | <0.0001 | <0.0001 | 0.998 |
| 4 | Petiole length (cm) | 0.000 | <0.0001 | 1.000 | 0.009 | <0.0001 | 1.000 |
| 5 | Days to flowering | <0.0001 | <0.0001 | 0.032 | <0.0001 | <0.0001 | 1.000 |
| 6 | Flower bud size (cm) | <0.0001 | <0.0001 | 0.078 | <0.0001 | <0.0001 | 0.899 |
| 7 | Number of flowers per raceme | <0.0001 | <0.0001 | 0.995 | <0.0001 | 0.000 | 0.003 |
| 8 | Peduncle length (cm) | 0.003 | <0.0001 | 1.000 | 0.003 | <0.0001 | 1.000 |
| 9 | Pods per peduncle | 0.742 | <0.0001 | 0.973 | <0.0001 | <0.0001 | 0.054 |
| 10 | Pod length (cm) | 0.194 | <0.0001 | 1.000 | 0.371 | <0.0001 | 1.000 |
| 11 | Pods per plant | <0.0001 | <0.0001 | <0.0001 | <0.0001 | <0.0001 | <0.0001 |
| 12 | Seeds per pod | 0.894 | <0.0001 | 1.000 | <0.0001 | <0.0001 | 0.712 |
| 13 | Seed size (mm$^2$) | <0.0001 | <0.0001 | 0.052 | 0.013 | <0.0001 | 0.506 |
| 14 | 100-Seed weight (g) | <0.0001 | <0.0001 | 0.037 | <0.0001 | <0.0001 | 0.068 |
| 15 | Yield (Kg/ha) | <0.0001 | <0.0001 | 0.032 | <0.0001 | <0.0001 | 0.055 |

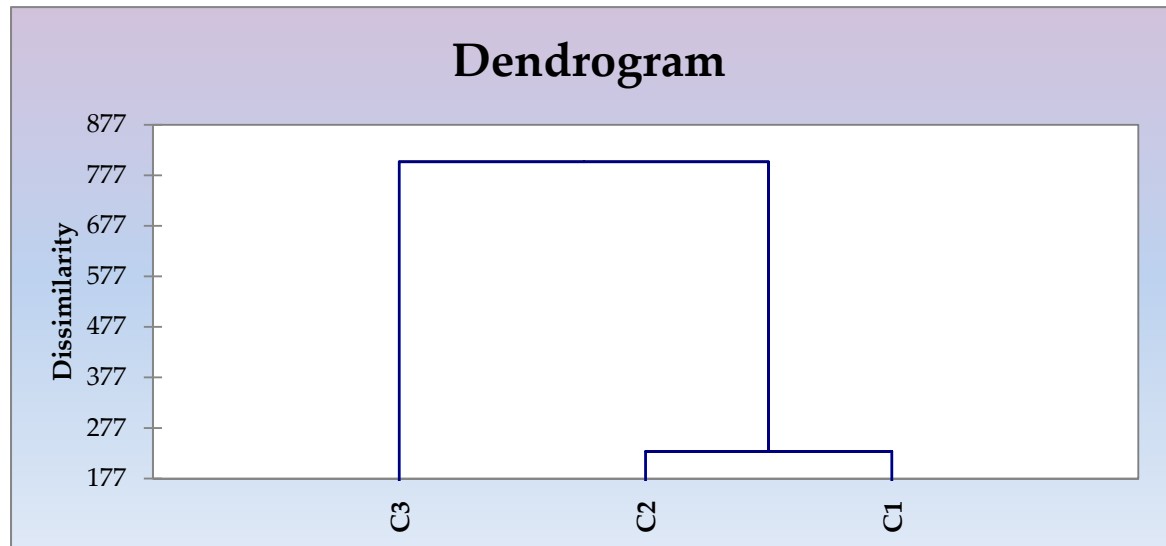

**Figure 6.** Dendrogram depicting the studied clusters of wild *Vigna* accessions for the 15 quantitative traits.

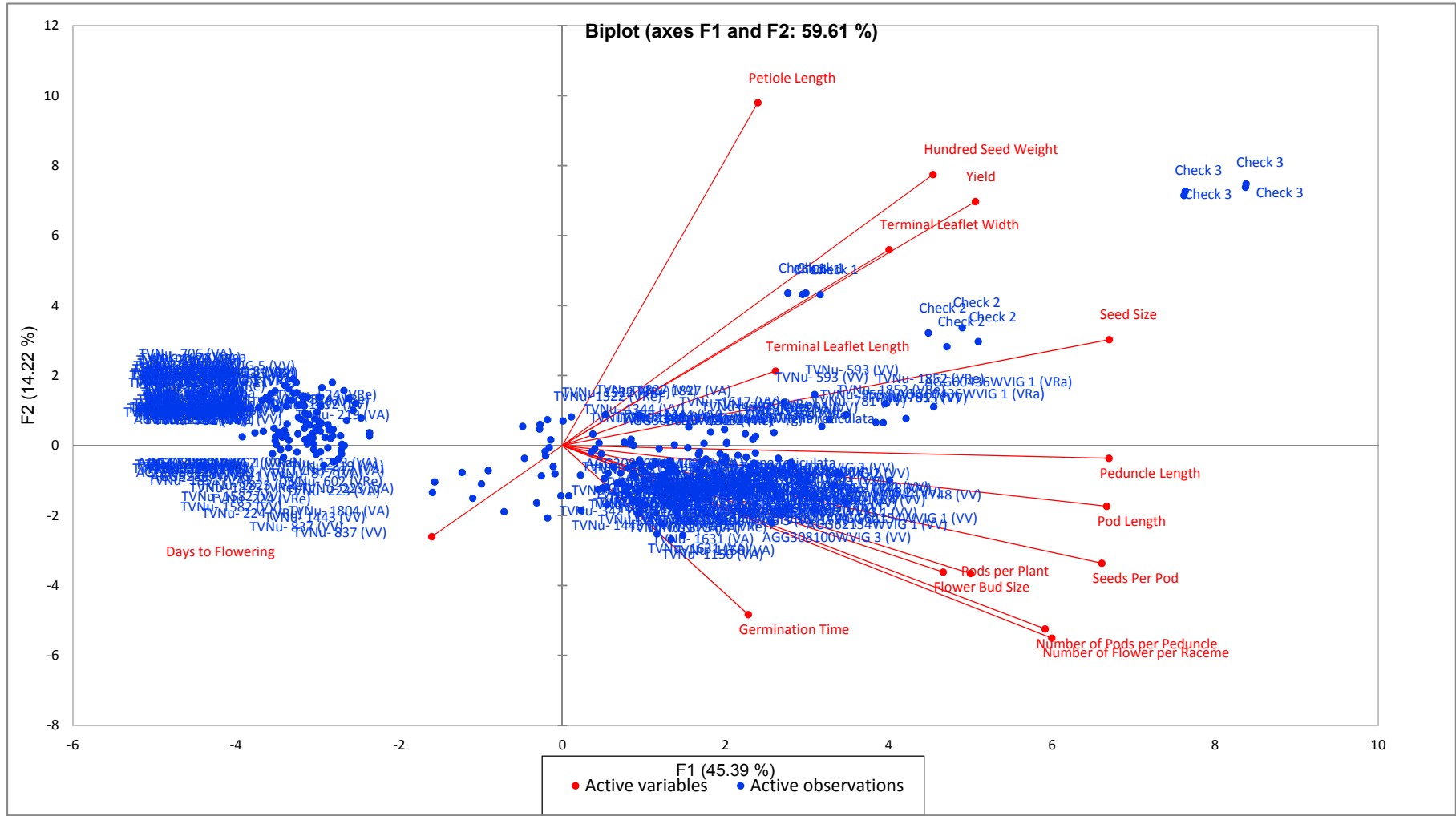

**Figure 7.** Principal component analysis result, showing the relationship between the traits and the accessions.

## 4. Discussion

The qualitative exploration of the wild *Vigna* species showed that there are variations in their characteristics for the same trait within the same species (Section 3.1), while all the checks expressed the same form of a particular trait throughout the experiments. Some of these qualitative characters are expressions of a genetic variation within the genome of the plant. A recent taxonomic differentiation was established between two wild Vigna species (*V. stipulacea* and *V. trilobata*) based on their morphological characteristics, such as germination habit, primary leaf attachment, etc. [14]. Therefore, the variations observed could be due to the heterogenous nature of the wild accessions, which have been homogenized in the checks through selection and breeding processes. It is a common opinion of many researchers that wild crop populations are much older and more diverse than domesticated crops, having undergone millennia of recombination, genetic drift, and natural selection [11]. Some of the trait forms found in the wild accessions might have only disappeared from the domesticated one during the domestication process. Some of the unique traits of the wild accessions, such as their leaf, stem, and petiole pubescence, which are not found in the checks, might have existed in those checks but disappeared with time during the domestication process. They could have a potential use, if they are domesticated, since they are thought to be responsible for some beneficial traits, such as the resistance to diseases and pests [15,16]. Therefore, it might be time to start examining some of the traits from the wild species that have disappeared in order to domesticate new species. The qualitative characteristics of the wild *Vigna* accessions found in this study were in line with most of the characteristics found in earlier works, carried out on other wild *Vigna* species [12,14].

Regarding the studied quantitative traits, Tables S2 and S3 summarized the means, ranges, and coefficients of variation at site A for only one season (the 2018 cropping season). This is due to the fact that, during the 2019 cropping season at site A, the rainfall was not enough (as per the pattern shown in Figure 1) to allow for germination and the growth of certain accessions. Most seeds did not germinate during that season, and those that did germinate (mainly checks) could not resist the harsh conditions. Figure 1 shows that the rainfall started at a very low rate (92.9 mm), then achieved its peak value (196.6 mm) and stopped. This amount of rainfall might have not been sufficient for the soil to allow the germination of the wild accessions. It is also known that the seed structure could influence the germination of seeds [17,18]. However, the characteristics of the seed structure of wild legumes are still yet to be reported.

The first cropping season at site A showed significant differences ($p < 0.05$) between the checks and the wild accessions for all the analyzed traits (Table 2a). Accession effects were found for all the traits, except for the number of flowers per raceme trait (trait 7) (Table 2a). This shows that the different accessions and species involved in the study possess different phenotypic and probably genetic characteristics. The number of flowers per raceme seemed to have no significant difference among the accessions and the checks. This might have been influenced by other agro-climatic conditions of the environment at that moment, which could affect some accessions, but probably not all. It has been reported that simple shading can affect the number of flowers per raceme [19]. The block effect observed at that site could be due to some particular factors of the field, ranging from agro-climatic to soil characteristics. The most probable explanation could be that the soil was heterogeneous in the same field and differently affected the checks. The ability of a plant to respond to soil characteristics can affect some of its physiological and phenotypic characteristics [20].

A similar pattern of results was observed during the two cropping seasons at site B. The observed phenomenon could have the same explanation as in the case of site A, mentioned above.

Based on the result shown in Table 5, only the days to flowering, pods per plant, hundred seed weight, and the yield were affected by their growing environment (accession × site effect), while only the number of flowers per raceme and the pods per plant were affected by the cropping season. These effects might be explained by the agro-climatic characteristics of each cropping site and season. As shown in Figure 1, site A has lower and shorter rainfall characteristics, which can affect the days to flowering. This is in line with earlier reports that predicted that the changes in the flowering time

are associated with a reduction in precipitation [21]. The effect of the yield and yield parameters, such as the pods per plant and hundred seed weight traits, has been reported before in relation to other legumes, and these reports do not contradict the present findings [22,23]. Therefore, it should be recommended that these traits be taken into consideration during any attempt to domesticate or improve wild legumes. The number of flowers per raceme and the pods per plant were the only traits affected by the cropping season. These two traits are closely related, as confirmed by the positive correlation that exists between the two, shown in Figure 7. They could also be directly or indirected affected by the variations in temperature and rainfall, as per the earlier explanation [20,21]. The significant effect of the season on the number of flowers per raceme and pods per plant has also been reported in relation to the landraces of *Phaseolus vulgaris* and cowpea (*Vigna unguiculata*) [24,25]. These two traits also need to be considered in any attempt at domestication.

Figure 6 revealed that the wild *Vigna* accessions could be grouped into three clusters, with one larger cluster (cluster I), including two checks (Table S4). This shows that some of the wild accessions share common features and probably genetic characteristics. Cluster I, containing the checks, could offer a clear orientation for the selection of candidates for domestication. Cluster 1 could also offer recommendations pertaining to the cooking time and water absorption capacity traits as reported earlier [26]. These are clear indications that these wild legumes could be domesticated and made useful, as the preliminary finding showed that farmers would be interested in utilizing them for various purposes [27]. In fact, it has recently been reported that Vigna stipulacea, another wild legume species with biotic resistance traits is domesticable [28] However, it is also necessary to note that domestication process could also affect the nutritional and health characteristics of the domesticated product as alerted by some researchers [29]. Therefore, the choice of *V. vexillata*, *V. reticulata*, *V. ambacensis*, and *V. racemosa* species in this study was first based on their availability in genebanks and from the little preliminary information obtained from the authors earlier investigations [5,26,27].

Figure 7 provides further indications relating to the domestication of these wild legumes by grouping them based on their quantitative agro-morphological traits. It was shown that most of the quantitative traits are positively correlated, and there is a degree of commonality between the checks and a group of some wild species.

## 5. Conclusions

This study revealed that the wild *Vigna* species possesses a large variation range of qualitative and quantitative traits, which could be exploited in the improvement of domesticated species or guide their domestication. Specifically, it was found that only the days to flowering, pods per plant, hundred seed weight and the yield were affected by their growing environment (accession x site effect), while only the number of flowers per raceme and the pods per plant were affected by the cropping season (accession x season effect). All the quantitative traits showed significant differences among accessions for each site and each season. The study further provides indications relating to *the* candidate accessions favorable for domestication, based on the quantitative and qualitative traits. However, further characterization, focusing on the biochemical content of the wild species, may be of great value for the extension of domestication information.

**Supplementary Materials:** The following are available online at http://www.mdpi.com/2073-4395/10/1/111/s1, Table S1: Wild Vigna legumes accessions used in the study, Table S2: Means, ranges and coefficients of variation for the selected quantitative traitsa, analyzed at site A* and B* during the two cropping seasons, Table S3: Adjusted mean values for selected quantitative traits per species at various sites, Table S4: Distribution of the accessions, according to the clusters generated from the agglomerative hierarchical analysis (AHC).

**Author Contributions:** P.A.N. and P.B.V. conceived and designed the experiments; D.V.H. performed the experiments, collected data, analyzed the data, and made the first draft of the manuscript; P.B.V. and A.O.M. supervised the research and internally reviewed the manuscript; and P.A.N. made the final internal review and revised the final draft of the manuscript. All authors have read and agreed to the published version of the manuscript.

**Funding:** This research was partially funded by the Centre for Research, Agricultural Advancement, Teaching Excellence and Sustainability in Food and Nutrition Security (CREATES-FNS) through the Nelson Mandela African Institution of Science and Technology (NM-AIST) and the World Bank. The research also received funding support from the International Foundation for Science (IFS) through Grant No. I-3-B-6203-1.

**Acknowledgments:** The authors are grateful to the Centre for Research, Agricultural Advancement, Teaching Excellence and Sustainability in Food and Nutrition Security (CREATES-FNS), as well as its partners, the Nelson Mandela African Institution of Science and Technology (NM-AIST), and the World Bank. The authors also acknowledge the additional funding support from the International Foundation for Science (IFS) through Grant No. I-3-B-6203-1. The authors are also grateful to the Genetic Resources Center, International Institute of Tropical Agriculture (IITA), Ibadan-Nigeria, as well as the Australian Grains Gene bank (AGG) for providing supporting information and seed materials for research.

**Conflicts of Interest:** The authors declare no conflict of interest.

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
