# Peer review of "Agro-Morphological Exploration of Some Unexplored Wild Vigna Legumes for Domestication"

_agronomy, doi:10.3390/agronomy10010111_

Round 1
Reviewer 1 Report
The authors have adequately addressed the concerns I raised in my previous review. The English is clear enough, and has been through a hired service, although I still find it somewhat Victorian in tone. The primary claims about the need for neodomestication have been properly adjusted, better addressing the significant limitations of a study with this design.
On line 66 of the first page of the introduction, there is no doubt that the Fabaceae are the third largest family. This sentence does not need a caveat.
Author Response
The authors would like to first thank you once again for your precious time and invaluable thoughtful comments towards the improvement of our article. We have carefully addressed all the comments requested. The corresponding changes and refinements made in the revised paper are summarized in our responses below. Generally, all the corrected points and changes made in the revised version of the manuscript can easily be tracked as the track changes option of Microsoft Word was activated during the revision. The changes inserted in the manuscript for this second round of revision upon the comments recommended by you, the reviewer #1 have been highlighted in pink color.
Comments and Suggestions for Authors
The authors have adequately addressed the concerns I raised in my previous review. The English is clear enough, and has been through a hired service, although I still find it somewhat Victorian in tone. The primary claims about the need for neodomestication have been properly adjusted, better addressing the significant limitations of a study with this design.
Response: Thank you, Sir. The remark is well noted.
Point 1: On line 66 of the first page of the introduction, there is no doubt that the Fabaceae are the third largest family. This sentence does not need a caveat.
Response 1: The sentence on line 66 has been rephrased to eliminate the warning tonality as recommended. Please, check page 2 (introduction section, blue highlight), line 66.

Reviewer 2 Report
The authors made changes according to the reviewer comments. However, due to a change in terminology, the same term is written twice. That is why the whole manuscript must be checked and corrected. For word checks these correction should be made on page page 4 line 141, page 14 lines 311 and 317, page 15 lines 327 and 337, page 28 line 445. For word accession the changes need to be made on 1 line 42, page 14 lines 312, 316, 322 and 326, page 15 lines 333, 337. 342 and 348, page 29 line 533. In case, some points are missed due to track changes, the authors are encouraged to make corrections throughout the manuscript.
Even though I find the word check a bit undefined, while the term cultivated accession is a well-defined term, the authors have chosen to use the word check for domesticated accession. This is explained for the first time in Materials and methods, thus on page 2 line 75 it should stay the term domesticated species and not checks.
Tables in the Results section should be self-explanatory, thus the authors should write the name of the location instead of calling it site A and B.
Specific suggestion:
Page 8 line 239 wild accessions and checksAuthor Response
The authors would like to first thank you once again for your precious time and invaluable thoughtful comments towards the improvement of our article. We have carefully addressed all the comments requested. The corresponding changes and refinements made in the revised paper are summarized in our responses below. Generally, all the corrected points and changes made in the revised version of the manuscript can easily be tracked as the track changes option of Microsoft Word was activated during the revision. The changes inserted in the manuscript for this second round of revision upon the comments recommended by you, reviewer #2 have been highlighted in pink color.
Comments and Suggestions for Authors
The authors made changes according to the reviewer comments.
Point 1: However, due to a change in terminology, the same term is written twice. That is why the whole manuscript must be checked and corrected.
Response 1: The whole manuscript has been checked and the terms or words written in double have corrected as recommended. All the corrected terms have been highlighted in pink color to ease tracking for your verification.
Point 2:
For word checks these correction should be made on page page 4 line 141, page 14 lines 311 and 317, page 15 lines 327 and 337, page 28 line 445.For word accession the changes need to be made on 1 line 42, page 14 lines 312, 316, 322 and 326, page 15 lines 333, 337. 342 and 348, page 29 line 533.
In case, some points are missed due to track changes, the authors are encouraged to make corrections throughout the manuscript.
Response 2:
The word <<checks>> has been corrected not to appear in double on the mentioned pages accordingly as recommended. Please check the following pages: page 4 line 141, page 14 lines 311, 316, and 326, page 15 lines 327 and 337, page 28 line 446. The word <<accession>> has been corrected not to appear in double on the mentioned pages accordingly as recommended. Please check the following pages: Page 1 line 42, page 14 lines 312, 316, 322 and 326, page 15 lines 333, 337. 341 and 347, page 29 line 534. The authors have checked all the missed points and corrected them. Page 15 line 332
Point 3: Even though I find the word check a bit undefined, while the term cultivated accession is a well-defined term, the authors have chosen to use the word check for domesticated accession. This is explained for the first time in Materials and methods, thus on page 2 line 75 it should stay the term domesticated species and not checks.
Response 3: The have replaced the term ‘’check’’ by ‘’domesticated species’’ on line 75 of page 2 as recommended. Thank you
Point 4: Tables in the Results section should be self-explanatory, thus the authors should write the name of the location instead of calling it site A and B.
Response 4: The names of the locations have been written on tables of the results section as recommended.
Specific suggestion:
Point 5: Page 8 line 239 wild accessions and checks
Response 5: The sentence of line 8 of page 8 has been corrected accordingly as pointed out.

This manuscript is a resubmission of an earlier submission. The following is a list of the peer review reports and author responses from that submission.
Round 1
Reviewer 1 Report
Harouna and colleagues have examined morphological variation in a number of uncultivated Vigna species, and interpreted their observations in the light of neodomesticating these species.
The field trial itself is a straightforward experiment, with two sites in an augmented randomized block design. This experiment is fine for measuring phenotypic variation, and is adequately described.
My primary concern with this manuscript is in the writing. Although mostly in proper English, it is far more Victorian than is common in other MDPI agronomy or other crop breeding papers. It needs a thorough re-writing, beyond the scope of reviewer comments. Professional editorial help is recommended to align the writing style with the journal.
I am also concerned about the interpretation. Although neodomestication is a fascinating topic, there is hardly unanimity across the crop breeding community in its importance. The abstract, introduction and discussion at the least should be rephrased to capture this lack of agreement. More importantly, the authors really have not measured anything about neodomestication. They have not examined what would need to be done to domesticate these wild species. For this to be a neodomestication paper, I think it needs more than simply measuring the wild species in a semi-cultivated setting. I think this is potentially a flaw in the manuscript. At the very least, in revisions the authors should dig more deeply into neodomestication.
Author Response
The authors would like to first thank the editors as well as the reviewers for your precious time and invaluable thoughtful comments towards the improvement of our article. We have carefully addressed all the comments requested. The corresponding changes and refinements made in the revised paper are summarized in our responses below. Generally, all the corrected points and changes made in the revised version of the manuscript can easily be tracked as the track changes option of Microsoft Word was activated during the revision. The changes inserted in the manuscript upon the comments recommended by reviewer #1 have been highlighted in Green while those inserted following reviewer #2 comments have been highlighted in yellow color.
REVIEWER #1 COMMENTS AND RESPONSES:
Comments and Suggestions for Authors
Harouna and colleagues have examined morphological variation in a number of uncultivated Vigna species, and interpreted their observations in the light of neodomesticating these species.
Point 1: The field trial itself is a straightforward experiment, with two sites in an augmented randomized block design. This experiment is fine for measuring phenotypic variation, and is adequately described.
Response 1: Thank you, Sir
Point 2: My primary concern with this manuscript is in the writing. Although mostly in proper English, it is far more Victorian than is common in other MDPI agronomy or other crop breeding papers. It needs a thorough re-writing, beyond the scope of reviewer comments. Professional editorial help is recommended to align the writing style with the journal.
Response 2: The manuscript was sent to the MDPI English Editing service for professional editing to align the manuscript with the journal style as recommended. The manuscript was registered under the editing service as <English editing ID: English-14465>.
Point 3: I am also concerned about the interpretation. Although neodomestication is a fascinating topic, there is hardly unanimity across the crop breeding community in its importance. The abstract, introduction and discussion at the least should be rephrased to capture this lack of agreement.
Response 3: The abstract, introduction and discussion parts have been improved to capture the lack of agreement on the importance of neo-domestication as recommended.
-For Abstract, check page 1, lines 22- 26.
- For Introduction, check page 2 lines 59- 63.
- For Discussion, check page 29, lines 514- 517.
Point 4: More importantly, the authors really have not measured anything about neodomestication. They have not examined what would need to be done to domesticate these wild species. For this to be a neodomestication paper, I think it needs more than simply measuring the wild species in a semi-cultivated setting. I think this is potentially a flaw in the manuscript. At the very least, in revisions the authors should dig more deeply into neodomestication.
Response 4: The authors agree with the reviewer that the paper is not describing a complete neo-domestication process but as stated in the topic, the paper is ‘’exploring’’ some specific agro-morphological characteristics of the wild legumes in order to expose some of the preliminary findings that could lead to a complete domestication process in the future.
These wild legumes species have very little information documented as stated in our previous review (Harouna et al., 2018). Some other few preliminary information on these wild species have recently been published and their domestication is still under studies. Some of the recent information on them can be found on the references below including the most recently published information about the domestication of Vigna stipulacea.
Harouna, D.V.; Venkataramana, P.B.; Ndakidemi, P.A.; Matemu, A.O. Under-exploited wild Vigna species potentials in human and animal nutrition: A review. Glob. Food Sec. 2018, 18, 1–11.
Harouna, D.V.; Venkataramana, P.B.; Matemu, A.O.; Ndakidemi, P.A. Assessment of Water Absorption Capacity and Cooking Time of Wild Under-Exploited Vigna Species towards their Domestication. Agronomy 2019, 9, 509.
Harouna; Venkataramana; Matemu; Ndakidemi Wild Vigna Legumes: Farmers’ Perceptions, Preferences, and Prospective Uses for Human Exploitation. Agronomy 2019, 9, 284.
Smýkal, P.; Nelson, M.N.; Berger, J.D.; Von Wettberg, E.J.B. The impact of genetic changes during crop domestication on healthy food development. Agronomy 2018, 8, 1–2.

Reviewer 2 Report
The work includes evaluation of some agro-morphological traits in unexploited wild Vigna species compared to some domesticated Vigna legumes.
The introduction is well written. The authors chose 4 wild species to analyse (out of 100 wild species in genus Vigna) due to low availability of any information. They analysed and identified traits that could be improved during domestication of the 4 species. However, if the authors are in possession of some information on why these 4 species have been chosen, it would further be of interest to the readers and improve the manuscript. There is only one sentence available in the discussion (page 29, lines 496-498), and if possible that part could be expanded.
Throughout the entire manuscript several terms are used to express the same word. One example would be: genotype, accession and treatment. This must be avoided as it brings confusion for the reader of the manuscript and makes the whole manuscript difficult to understand. Similarly, domesticated species are also called checks and controls, thus adding more to the confusion. The authors are thus encouraged to choose a term and use it throughout the entire manuscript.
In Materials and methods section (2.1. Sample Collection and Preparation) it is said that 160 accessions (taken from gene banks or self-collected) were analysed and the reference is made in the Table 1, in which there are only 84 accessions collected. Therefore, I recommend that the authors make a table naming all accession IDs, species they belong to and the gene bank they were obtained from. This table could be included in the Supplement. The authors should write the Accession IDs of the domesticated species collected from gene bank in the sub-section 2.1. Sample Collection and Preparation.
In the section Results, all the Figures and Tables are put at the end of the corresponding sub-section. A suggestion to the authors would be to choose the most representative tables and figures and put them in the appropriate place in the manuscript, while the rest should be published in the Supplement, thus contributing to the continuation and the comprehensiveness of the manuscript.
In the Table 9., there are 138 accessions named (excluding domesticated species that the authors called checks). This should be addressed in the manuscript.
Some specific suggestions are following:
Page 1, line 54 – United Nations (capital U in United) Page 1, line 83 - please add comma after non-domesticated Vigna species Page 1, line 87 - based on... Page 2, line 91 - That later on... Page 4, line 143 – please write full name of IPGRIAuthor Response
The authors would like to first thank the Editors as well as the reviewers for your precious time and invaluable thoughtful comments towards the improvement of our article. We have carefully addressed all the comments requested. The corresponding changes and refinements made in the revised paper are summarized in our responses below. Generally, all the corrected points and changes made in the revised version of the manuscript can easily be tracked as the track changes option of Microsoft Word was activated during the revision. The changes inserted in the manuscript upon the comments recommended by reviewer #1 have been highlighted in Green while those inserted following reviewer #2 comments have been highlighted in yellow color.
REVIEWER #1 COMMENTS AND RESPONSES:
Comments and Suggestions for Authors
The work includes evaluation of some agro-morphological traits in unexploited wild Vigna species compared to some domesticated Vigna legumes.
Point 1: The introduction is well written. The authors chose 4 wild species to analyse (out of 100 wild species in genus Vigna) due to low availability of any information. They analysed and identified traits that could be improved during domestication of the 4 species. However, if the authors are in possession of some information on why these 4 species have been chosen, it would further be of interest to the readers and improve the manuscript. There is only one sentence available in the discussion (page 29, lines 496-498), and if possible that part could be expanded.
Response 1: More information on the choice of the studied species has been added as recommended. Please, check page 29, lines 517- 520.
Point 2: Throughout the entire manuscript several terms are used to express the same word. One example would be: genotype, accession and treatment. This must be avoided as it brings confusion for the reader of the manuscript and makes the whole manuscript difficult to understand. Similarly, domesticated species are also called checks and controls, thus adding more to the confusion. The authors are thus encouraged to choose a term and use it throughout the entire manuscript.
Response 2: The authors have made the necessary adjustment of words as recommended. The words ‘’genotype’’ and ‘’treatment’’ have been replaced by ‘’accession’’ while the words ‘’domesticated species’’ and ‘’control’’ have been changed to ‘’checks’’ throughout the manuscript. This can be followed through the track changes of the word.
Point 3: In Materials and methods section (2.1. Sample Collection and Preparation) it is said that 160 accessions (taken from gene banks or self-collected) were analysed and the reference is made in the Table 1, in which there are only 84 accessions collected. Therefore, I recommend that the authors make a table naming all accession IDs, species they belong to and the gene bank they were obtained from. This table could be included in the Supplement. The authors should write the Accession IDs of the domesticated species collected from gene bank in the sub-section 2.1. Sample Collection and Preparation.
Response 3: The authors have improved the Table 1 as Table S1 and added it as a supplementary material as suggested. Please, check page 3-4 of the revised manuscript and supplementary material Table S1 to confirm.
Point 4: In the section Results, all the Figures and Tables are put at the end of the corresponding sub-section. A suggestion to the authors would be to choose the most representative tables and figures and put them in the appropriate place in the manuscript, while the rest should be published in the Supplement, thus contributing to the continuation and the comprehensiveness of the manuscript.
Response 4: The authors have chosen the relevant Tables to remain in the manuscript and have attached other tables as supplementary material as recommended by the reviewer. Tables 1, 3, 4 and 9 have been removed from the manuscript as supplementary materials and renamed as Tables S1, S2, S3 and S4. The necessary adjustment of their references in the text has also been done accordingly. Check pages 3, 16, 17, 25 and supplementary materials (Tables S1, S2, S3 and S4) for confirmation.
Point 5: In the Table 9., there are 138 accessions named (excluding domesticated species that the authors called checks). This should be addressed in the manuscript.
Response 5: The raised issue has been addressed accordingly. Check page 15, lines 349- 352.
Some specific suggestions are following:
Point 6: Page 1, line 54 – United Nations (capital U in United)
Response 6: The mistake has been corrected accordingly, please check page 2, line 56 to confirm.
Point 7: Page 1, line 83 - please add comma after non-domesticated Vigna species
Response 7: The comma has been added as recommended, check page 2, line 87 to confirm.
Point 8: Page 1, line 87 - based on...
Response 8: The typographical error has been corrected accordingly as pointed out. Check page 3, line 91 to confirm.
Point 9: Page 2, line 91 - That later on...
Response 9: The pointed error has been corrected, see page 3, line 95.
Point 10: Page 4, line 143 – please write full name of IPGRI
Response 10: the full name of IPGRI has been added as recommended. Please check page 4, line 148 to confirm.
